ecology, plant science

ecosystem productivity, individual plant growth, *n*-dimensional hypervolume concept, optical types, spectral types, spectroscopy

**Author for correspondence:**
Anna K. Schweiger
e-mail: anna.k.schweiger@gmail.com

# Coupling spectral and resource-use complementarity in experimental grassland and forest communities

Anna K. Schweiger[1,2,3], Jeannine Cavender-Bares[1,4], Shan Kothari[3,4], Philip A. Townsend[5], Michael D. Madritch[6], Jake J. Grossman[7,8], Hamed Gholizadeh[9], Ran Wang[10] and John A. Gamon[10,11]

[1]Department of Ecology, Evolution and Behavior, University of Minnesota, Saint Paul, MN, USA
[2]Remote Sensing Laboratories, Department of Geography, University of Zurich, Zurich, Switzerland
[3]Institut de recherche en biologie végétale and département de sciences biologiques, Université de Montréal, Montréal, Québec, Canada
[4]Department of Plant and Microbial Biology, University of Minnesota, Saint Paul, MN 55108, USA
[5]Department of Forest and Wildlife Ecology, University of Wisconsin-Madison, Madison, WI, USA
[6]Department of Biology, Appalachian State University, Boone, NC, USA
[7]Biology Department, Swarthmore College, Swarthmore, PA, USA
[8]Arnold Arboretum of Harvard University, Boston, MA, USA
[9]Center for Applications of Remote Sensing, Department of Geography, Oklahoma State University, Stillwater, OK, USA
[10]Center for Advanced Land Management Information Technologies (CALMIT), School of Natural Resources, University of Nebraska-Lincoln, Lincoln, NE, USA
[11]Departments of Earth and Atmospheric Sciences and Biological Sciences, University of Alberta, Edmonton, Alberta, Canada

AKS, 0000-0002-5567-4200; JC-B, 0000-0003-3375-9630; SK, 0000-0001-9445-5548; PAT, 0000-0001-7003-8774; JJG, 0000-0001-6468-8551; HG, 0000-0002-4770-7893; RW, 0000-0002-3810-9103; JAG, 0000-0002-8269-7723

Reflectance spectra provide integrative measures of plant phenotypes by capturing chemical, morphological, anatomical and architectural trait information. Here, we investigate the linkages between plant spectral variation, and spectral and resource-use complementarity that contribute to ecosystem productivity. In both a forest and prairie grassland diversity experiment, we delineated *n*-dimensional hypervolumes using wavelength bands of reflectance spectra to test the association between the spectral space occupied by individual plants and their growth, as well as between the spectral space occupied by plant communities and ecosystem productivity. We show that the spectral space occupied by individuals increased with their growth, and the spectral space occupied by plant communities increased with ecosystem productivity. Furthermore, ecosystem productivity was better explained by inter-individual spectral complementarity than by the large spectral space occupied by productive individuals. Our results indicate that spectral hypervolumes of plants can reflect ecological strategies that shape community composition and ecosystem function, and that spectral complementarity can reveal resource-use complementarity.

## 1. Introduction

Plants partition resources in space and time as a result of contrasting ecological strategies and evolutionary histories, giving rise to biochemical, structural or phenological differences that determine the optical properties of leaves and canopies. Spectral profiles of plants, defined here as the reflectance spectra of plant leaves or whole plants at high-spectral resolution, are influenced by leaf traits [1–5], including pigment composition, micro- and macronutrient content, water content, specific leaf area (SLA), leaf surface properties and leaf internal

structure. When measured remotely, plant spectra are also influenced by biophysical parameters of vegetation—whole-plant traits related to canopy architecture, including leaf area index, leaf angle distribution, canopy height and canopy cover [2,6,7]. Consequently, spectral profiles capture key differences in foliar chemistry, leaf anatomy, morphology, life-history strategies and responses to environmental variation [4,5], which have consequences for ecosystem structure and function above- and belowground [5,8].

Optical types are functional differences among plants captured through optical techniques, of which spectral types represent a subset. The optical type concept [9], published over a decade ago, posits that since neighbouring plants share resources, including light, nutrients and water, the degree of spectral complementarity in plant communities can inform us about complementary resource-use, which is, in turn, associated with ecosystem function and productivity. Complementary light use strategies have since been studied using spectral data in tropical forests [10], among evergreen and deciduous trees [11], and among prairie species [12]. Other studies have shown that dissimilarity in spectral profiles of plants correlates with their functional dissimilarity [13–17] and evolutionary divergence time [14,15,18,19]; and positive relationships between measures of spectral diversity and eco-system productivity [15] suggest that spectral differences among plants are coupled to resource-use complementarity. However, this central aspect of the optical type concept has not yet been exhaustively tested.

Here, we examine the concept of the optical types using $n$-dimensional spectral hypervolumes to explore the relationship between spectral and resource-use complementarity. We define the 'spectral hypervolume' as the $n$-dimensional space occupied by plants and delineated by spectral axes—spectral bands or other expressions of spectral variation—along which plants can vary. Plant spectral hypervolumes can be quantified across biological scales from individual plants to species, lineages or even plant communities. The spectral hypervolume occupied by an individual plant can be quantified by leaf spectra from that plant. Its size represents intra-individual spectral variation [20] and is conjectured to be indicative of the range of environmental conditions experienced by the plant, including gradients of light, wind and temperature, as well as variation in pathogen and herbivore pressure within the canopy. Greater variation in environment and leaf spectra within a plant give rise to a larger spectral hypervolume, which is in turn, we hypothesize, positively correlated with plant growth and biomass. The spectral hypervolume occupied by a species can be quantified by spectral profiles of individuals from that species within a certain ecosystem or distributed across ecosystems. The size of species' spectral hypervolumes represents intraspecific spectral variation. Distances among the spectral hypervolumes of different species are correlated with species' spectral dissimilarities. Moreover, the degree of overlap among species' spectral hypervolumes is indicative of their shared spectral characteristics, which are associated with shared functional attributes [2,9,15] and shared ancestry [5,14,15,19]. The spectral hypervolume occupied by a plant community can be quantified by the spectral profiles of individuals and species within that community. The size of the spectral hypervolume occupied by a plant community represents the variation among spectral types within that community, or its spectral diversity, which in turn is hypothesized to

indicate the community's taxonomic, functional and phylo-genetic diversity [15,21]. Greater variation among plant spectra within a community gives rise to a larger community hypervolume. Overlaps among the spectral hypervolumes of plant communities indicate similarity in resource-use patterns among these communities, while distances among spectral hypervolumes of plant communities are hypothesized to predict the degree of dissimilarity in resource-use patterns among them and the dissimilarity in environmental conditions they experience.

We define 'spectral complementarity' as the separation of spectral hypervolumes of plants within multidimensional spectral space. Spectrally complementary plants, which represent contrasting spectral types, can be assumed to use resources, including light, nutrients and water, differently. This is because resource-use strategies of plants are reflected by their foliar chemistry, anatomy and morphology [22,23], all of which influence the spectral response following the physics of light absorption and scattering [2]. From an individual perspective, spectral complementarity means that leaves from different parts of the canopy complement each other in terms of resource-use, for example, through foliar adaptations in response to light gradients within canopies [20,24]. From a stand or community perspective, spectral complementarity means that different individuals or species partition resources which allows them to compete less with each other and use the total resource pool together more completely [25–27] leading to a positive relationship between spectral complementarity, total resource-use and productivity.

The framework of the $n$-dimensional hypervolume has a long history in ecology. Proposed by Hutchinson [28] in connection with the $n$-dimensional niche concept, hyper-volumes have been used frequently to quantify the multivariate environmental or trait space occupied by organisms, including plants [29], and to describe the niches or multivariate functions of species or broader clades. Hypervolumes have also been used to describe the geographical range in which species and communities of organisms occur [30,31]. Notably, in remote sensing, there are well-established analogous concepts, including spectral endmembers, that are related to the positioning of reflectance spectra in multi-dimensional spectral space which are frequently used for feature extraction and classification [32]. The basic notion is that to differentiate and classify objects from remotely sensed imagery into groups, these groups need to show sufficient separation in spectral space for distinct clusters to emerge.

Here, we use spectral hypervolumes to assess variation among plants at two biological scales—at the individual and community scale—to understand the consequences of these dissimilarities for ecosystem function. Our overarching proposition is that the spectral complementarity of plants is indicative of their ecological complementarity in terms of resource use. Based on this idea, we test two hypotheses using data collected in two experiments, forest and biodiversity I (FAB) [33]—a tree diversity experiment (figure 1a)—and biodiversity II (BioDIV)—a prairie grassland diversity experiment (figure 1b) [34]. First, we hypothesize that greater spectral variation among leaves within focal plants, which is expected to result in larger individual spectral hypervolumes, will be associated with greater individual growth (FAB). Second, we hypothesize that plant communities that occupy greater total spectral space—either due to spectral complementarity among species or due to individuals that occupy

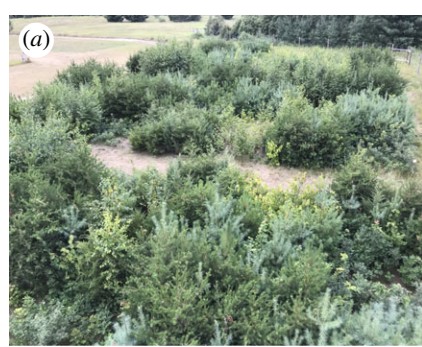
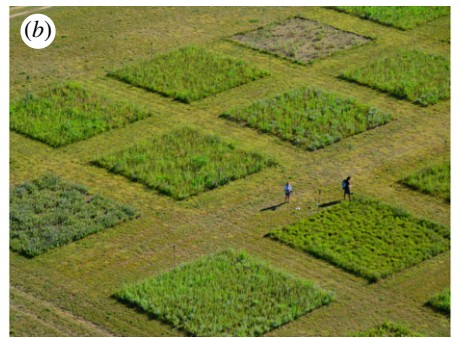

**Figure 1.** We tested the overarching hypothesis that spectral complementarity of plants is indicative of their resource-use complementarity in two experiments at the Cedar Creek Ecosystem Science Reserve (East Bethel, MN): (*a*) the forest and biodiversity I (FAB)—a tree diversity experiment—and (*b*) biodiversity II (BioDIV)—a prairie grassland diversity experiment. Photos by (*a*) Jeannine Cavender-Bares and (*b*) Jacob Miller. (Online version in colour.)

large spectral hypervolumes or both—will be associated with more productive ecosystems (FAB and BioDIV).

## 2. Methods

### (a) Study sites

Both FAB [33], which is part of IDENT (the International Diversity Experiment Network with Trees), and BioDIV [34] are located at the Cedar Creek Ecosystem Science Reserve (East Bethel, MN). We used leaf spectra, plant height and diameter measurements of 537 individuals from 12 tree species sampled in 68 plots in FAB in July 2016; and leaf spectra of 902 individuals from 14 grassland–savannah perennials sampled in 35 plots in BioDIV and aboveground biomass determined in the same plots in July 2015. The species sampled in FAB were: *Acer negundo* L. (30 ind.), *Acer rubrum* L. (47 ind.), *Betula papyrifera* Marshall (44 ind.), *Juniperus virginiana* L. (39 ind.), *Pinus banksiana* Lamb. (47 ind.), *Pinus resinosa* Aiton (52 ind.), *Pinus strobus* L. (47 ind.), *Quercus alba* L. (42 ind.), *Quercus ellipsoidalis* E. J. Hill (49 ind.), *Quercus macrocarpa* Michx. (50 ind.), *Quercus rubra* L. (39 ind.) and *Tilia americana* L. (51 ind.). The species sampled in BioDIV were: *Achillea millefolium* L. (49 ind.), *Amorpha canescens* Pursh (28 ind.), *Andropogon gerardii* Vitman (162 ind.), *Asclepias tuberosa* L. (70 ind.), *Lespedeza capitata* Michx. (99 ind.), *Liatris aspera* Michx. (49 ind.), *Lupinus perennis* L. (121 ind.), *Panicum virgatum* L. (49 ind.), *Petalostemum candidum* (Willd.) Michx. (28 ind.), *Petalostemum purpureum* (Vent.) Rydb. (52 ind.), *Petalostemum villosum* Nutt. (42 ind.), *Schizachyrium scoparium* (Michx.) Nash (76 ind.), *Solidago rigida* L. (50 ind.) and *Sorghastrum nutans* (L.) Nash (27 ind.).

### (b) Spectral data

We measured leaf spectra using two portable field spectrometers (SVC HR-1024i, Spectra Vista Corp., Poughkeepsie, NY; and PSR+ 3500, Spectral Evolution Inc., Lawrence, MA) and the associated leaf-clip assemblies (LC-RP PRO, Spectra Vista Corp.; and PSR+ 3500 leaf clip, Spectral Evolution Inc.). We used the SVC instrument covering the wavelength range from 340.5 to 2522.8 nm in 1024 spectral bands, and the PSR+ instrument covering the wavelength range from 350 to 2500 nm in 2151 spectral bands for spectral measurement of herbaceous and tree species, respectively. To characterize one individual spectrally, we measured the reflectance of either three or five mature, healthy leaves per individual depending on plant height. We measured three leaves for individuals under 30 cm in height—two from the top and one from the bottom-canopy layer—and five leaves for individuals over 30 cm in height—two from the top-, two from the mid- and one from the bottom-canopy layer.

In FAB, we sampled three individuals per species and for each individual we measured five leaves. We averaged measurements per canopy layer (top, mid and bottom), resulting in nine spectra from three individuals in monocultures, 18 spectra from six individuals in bi-cultures, 45 spectra from 15 individuals in five-species plots and 108 spectra from 36 individuals in 12-species plots. In BioDIV, we divided each plot into nine equally sized subplots and collected spectral data in four to eight subplots, depending on species richness; the centre subplot was always excluded to prevent disturbance. In mono- and bi-cultures, we sampled the four corner subplots; in four-species plots, we sampled the four corner and two additional outer subplots; in eight- or 16-species plots, we sampled all eight outer subplots. We measured four randomly selected individuals per subplot, making sure all species per plot were included (for details see [15]). Most individuals in BioDIV were under 30 cm and showed little intra-individual spectral variation. We thus averaged spectra per individual, resulting in 16 spectra in mono- and bi-cultures, 24 spectra in four-species plots and 32 spectra in eight- and 16-species plots.

Spectra were automatically corrected for dark current and stray light, and referenced against the Spectralon 99% reflectance standard disc (Labsphere, North Sutton, NH) of the leaf clip approximately every 10 min. Spectral data processing included removing mismeasurements, correcting discontinuities at the sensor overlap regions (between the Si and first InGaAs sensors, around 1000 nm, and between the first and second InGaAs sensors, around 1900 nm), removing noisy regions at the beginning and end of the spectrum, and interpolating spectra to 1 nm resolution. For spectral processing, we used the spectrolab [35] package in R [36].

### (c) Measures of individual plant growth and community productivity

In the FAB experiment, we measured the tree height (cm) and basal diameter (mm) for each tree. Trees in FAB were planted at the same time and at a similar size such that plant size provided a measure of plant biomass closely related to growth rate and productivity (for details see [33]). Aboveground net primary productivity was determined from increments in allometrically derived stem biomass (kg yr$^{-1}$). From this estimate, we calculated overyielding, the excess biomass produced by mixed-species plots compared to what would be expected based on monoculture yields, as a measure of the net biodiversity effect (NBE). In addition, we partitioned the NBE into complementarity (CE) and selection effects (SEs), following Loreau & Hector [27]. In the BioDIV experiment, we used biomass (g m$^{-2}$, dry weight) determined in clip strips as a measure of aboveground net primary productivity. Individual plant growth was not measured in BioDIV. Also, we did not calculate and partition the NBE in BioDIV because monocultures are not replicated in this

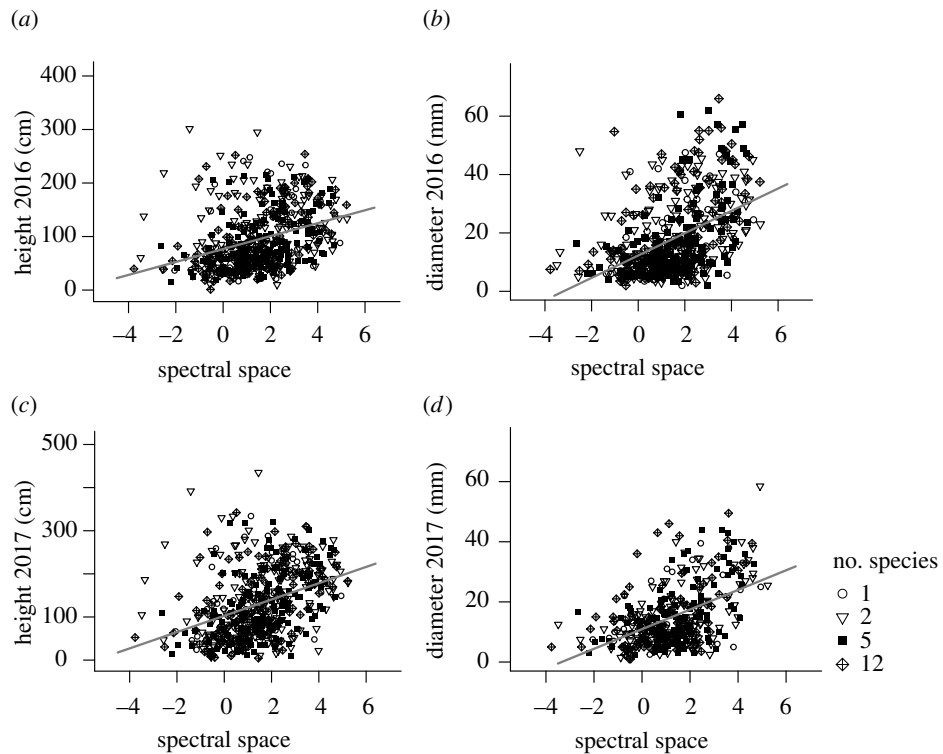

**Figure 2.** The spectral space occupied by individual trees in the FAB experiment predicts (*a,c*) tree height (2016: $n = 528$, $r^2 = 0.11$, $F_{1,526} = 65.3$, $p < 0.001$; 2017: $n = 520$, $r^2 = 0.14$, $F_{1,518} = 84.4$, $p < 0.001$) and (*b,d*) stem diameter (2016: $n = 528$, $r^2 = 0.21$, $F_{1,526} = 143.5$, $p < 0.001$; 2017: $n = 394$, $F_{1,392} = 124.0$, $p < 0.001$). The number of species planted per plot is indicated with different symbols; spectral hypervolume sizes are log-transformed.

experiment and calculations of the NBE and its components depend on estimates of mean monoculture yields.

### (d) Spectral space occupied by individuals and plant communities

We tested the degree to which the spectral space occupied by individual plants predicts plant growth by fitting regression models between the spectral space occupied by individual trees sampled in FAB and tree height (cm) or basal diameter (mm), which served as proxies for biomass. We reduced the dimensionality of spectral data by using the first three principal component (PC) axes, which explained more than 98% of the total spectral variation. We calculated the spectral space occupied by each individual using the R package hypervolume [37] based on Gaussian kernel density estimation and a Silverman bandwidth estimator with a quantile threshold of 5%.

Next, we tested the degree to which the spectral space occupied by plant communities predicts aboveground productivity. Since hypervolume size is known to be positively correlated with sample size [29], we used nine randomly selected spectral measurements per plot to calculate the spectral space occupied by plant communities in FAB, resulting in a total of 67 communities used for analysis. We iterated the random selection process 50 times and calculated the average and standard deviation of the spectral hypervolume size across iterates for each community. In BioDIV, we used 12 randomly selected individuals per plot—three random individuals from four random subplots per plot—resulting in a total of 30 communities used for analysis. Again, we ran 50 iterations of the selection process and averaged spectral hypervolume size per community. We reduced data dimensionality to the first three PC axes, which explained more than 98% of the total spectral variation and calculated the spectral space occupied per community using the R package hypervolume [37] using the same settings as above. Then, we tested the association between the spectral space occupied by plant communities and community productivity using linear

regression models. In FAB, we also tested the degree to which CE and SE were associated with the size of the spectral space occupied by plant communities. In addition, we tested in both experiments whether the spectral space occupied by plant communities increases with species richness.

## 3. Results

### (a) Individual hypervolume size increases with plant biomass

The spectral space occupied by individual trees—a measure of intra-individual spectral variation—was positively correlated with tree height and diameter (figure 2), both of which predict biomass and are closely correlated with productivity—the increase in biomass per unit time. This is probably because trees that grow more tend to have larger, more complex canopies and higher foliar plasticity than trees that grow less. The spectral space occupied by individuals was more closely associated with tree diameter than tree height. One explanation is that once trees are taller than their neighbours, it may be more advantageous to invest in mechanical stability and horizontal canopy extension than in vertical growth to maximize light interception.

### (b) More productive communities occupy larger spectral hypervolumes than less productive communities

The spectral space occupied by plant communities remained remarkably stable across iterated hypervolume calculations, in both FAB (standard deviations from 0.0054 to 0.0083 for log(hypervolumes) between 1.04 and 4.54) and BioDIV (standard deviations from 0.0052 to 0.0072 for log(hypervolumes) between 0.48 and 3.39). We thus based further analysis on the

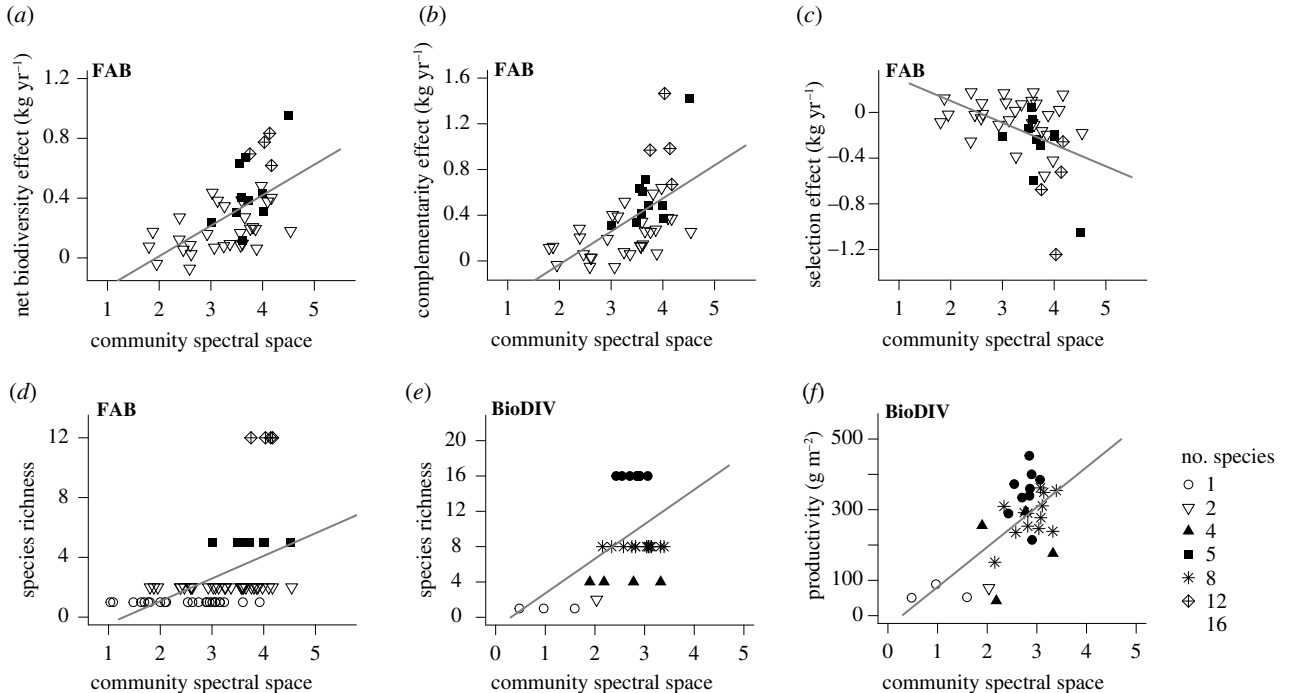

**Figure 3.** The spectral space occupied by plant communities predicts (*a*) the net biodiversity effect (NBE, $n = 43$, $r^2 = 0.34$, $F_{1,41} = 20.7$, $p < 0.001$), as well as (*b*) complementarity (CE, $n = 43$, $r^2 = 0.33$, $F_{1,41} = 20.4$, $p < 0.001$) and (*c*) selection effects (SE, $n = 43$, $r^2 = 0.19$, $F_{1,41} = 9.4$, $p < 0.004$) in the FAB experiment. The spectral space occupied by plant communities is also positively correlated with species richness in (*d*) FAB ($n = 67$, $r^2 = 0.23$, $F_{1,65} = 18.9$, $p < 0.001$) and BioDIV ($n = 30$, $r^2 = 0.25$, $F_{1,28} = 9.4$, $p < 0.005$); and it predicts (*f*) aboveground productivity in the BioDIV experiment ($n = 30$, $r^2 = 0.46$, $F_{1,28} = 24.1$, $p < 0.001$). The number of species planted per plot is indicated with different symbols; spectral hypervolume sizes are averaged across 50 resampling iterations and log-transformed.

average spectral space occupied per community. The spectral space occupied by tree communities in FAB explained 34% of overyielding (figure 3*a*) and increased with the number of species per community (figure 3*d*). Partitioning the NBE into its two components, CE and SE, revealed a positive relationship between the spectral space occupied by communities and complementarity (figure 3*b*), while the association with the SE was negative (figure 3*c*). In other words, compared to less productive communities, more productive communities did not, on average, harbour more highly productive individuals which occupy larger spectral spaces, but rather more spectrally complementary species that collectively contributed to the large spectral space occupied by these communities. Similarly, the spectral space occupied by communities in BioDIV increased with the number of species per plant community (figure 3*e*) and explained 46% of the total variation in aboveground productivity (figure 3*f*).

## 4. Discussion

Plant spectra provide integrative measures of plant phenotypes. Quantification of plant *n*-dimensional spectral hypervolumes offers a novel and effective way to assess variation among plants associated with resource-use complementarity that is predictive of plant growth and ecosystem productivity. We find that greater spectral complementarity—the degree to which individuals and species occupy distinct hypervolumes in spectral space—is associated with greater resource capture and growth in focal plants, and greater ecosystem productivity in communities with larger spectral hypervolumes. Consequently, we posit that spectral complementarity provides a measure of resource partitioning in plant communities. A series of functional traits of plants,

including traits specific to organs that interact minimally with light, such as roots and seeds, may not influence the spectral signal directly. However, plant spectra integrate many aspects of plant form and function, including biochemical, anatomical and morphological traits [5,9], which are sometimes expressed in coordination across the whole plant [38]. For example, as plants use light to power photosynthesis, their physiology responds in a coordinated way to solar radiation and atmospheric conditions (e.g. vapour pressure deficit), and also to belowground resources (e.g. water and nutrients). This leads to responses throughout the canopy including short- and mid-term changes in pigment pool sizes [16], hydraulic properties [39] and adaptations in leaf structure, the leading axes of spectral variation [40], providing a strong basis for using spectra as measures of plant function.

## (a) Spectral hypervolumes occupied by individuals and plant communities

The positive relationships between intra-individual spectral variation and tree height and diameter (figure 2) confirm our hypothesis that plants occupying more spectral space have greater growth. Higher growth rates create stronger light gradients within canopies driving plasticity in foliar traits, including SLA, pigments and nitrogen content [41,42], which all influence the spectral response. Foliar plasticity allows trees to minimize the costs of light capture in internal leaves relative to the benefits, which makes light capture more efficient overall than if there were no plasticity under the same conditions. Over time, taller trees seem to be able to sustain and perhaps even increase the benefits gained from harnessing a more diverse light environment (figure 2*c,d*), pointing towards size-asymmetric (size-dependent) light competition [43] among the trees in the FAB experiment.

More productive plant communities occupied larger spectral hypervolumes in the FAB and BioDIV experiments (figure 3*a,f*). However, while tree growth in FAB was positively correlated with intra-individual spectral variation (figure 2), the size of the spectral hypervolumes occupied by productive individuals did not explain the positive association between community productivity and the size of the spectral hypervolume occupied by these communities (figure 3*c*). In our case, it appears that the size of the spectral hypervolume occupied by productive plant communities is largely attained by spectral complementarity (figure 3*b*), which also contributes to contrasting spectral patterns among evergreen and deciduous boreal trees [11], prairie species [12] and species in dry tropical forests [16]. We posit that the large spectral space occupied by spectrally diverse communities can be explained by positive feedback, whereby high spectral dissimilarity is both a consequence of—and results in—greater resource capture and increased investment compared to spectrally depauperate communities. Given that more spectrally dissimilar species tend to be more functionally dissimilar [13,15], we interpret the total spectral space occupied by a plant community as a measure of its functional complexity and diversity.

## (b) Future applications of plant spectral hypervolumes

Investigating changes in the size, shape, overlap and position of spectral hypervolumes of individual plants, species and communities as they respond to a changing environment provide rich avenues for further exploration of the linkages between spectroscopy and ecological theory. Identifying the key spectral features and related anatomical, morphological and architectural characteristics that contribute most to these shifts in spectral hyperspace can indicate temporal and spatial resource partitioning [44]. Empty volumes (holes) in the spectral space occupied by plant communities might indicate colonization or invasion potential. Alternatively, they might indicate biologically unrealized spectral types [45], because biophysical tradeoffs in plant functional traits, such as between nitrogen content or photosynthetic capacity and SLA [22], limit the degree of spectral variation possible. These limitations might cause species' spectra to diverge or converge in particular cases, and may also cause species or functional groups to follow unique trajectories in spectral space in response to varying environmental conditions and phenology. Further research is needed to investigate the degree to which such changes in spectral hypervolumes might be useful for detecting the environmental change in its early stages. In addition, incorporating spectral hypervolume shifts through time in plant identification models would allow plant taxa that are spectrally similar at one point in time to be differentiated with time-series data. Such an approach would be particularly useful in diverse environments and for large-scale studies using remote sensing data.

## (c) Conclusions

Here, we propose the measurement of spectral hypervolumes—or the *n*-dimensional spectral space occupied by plants—as a novel means to investigate plant differentiation associated with complementarity in resource-use by individuals and within whole communities. This method builds upon a rich history of ecological theory related to resource-use and functional diversity. The ecological value of spectral hypervolumes stems from the fact that plants' reflectance spectra integrate many important dimensions of plant form and function [5,15], including biochemical, anatomical and morphological characteristics, that are related to resource capture and investment. We found that the growth of focal plants was associated with the size of the spectral hypervolume occupied by these individuals and that ecosystem aboveground productivity was associated with the size of the spectral hypervolume occupied by a plant community. Notably, ecosystem productivity was predominantly explained by spectral complementarity—the multidimensional separation of plants in spectral space—and not by the large spectral hypervolumes occupied by productive individuals. Spectral hypervolumes unite ecological theory with the physics of light capture, demonstrating the potential to reveal plant–environment interactions and resource partitioning over large areas using rapid field optical sampling or suitable remote sensing methods. Key to implementing this approach across large spatial extents will be developing scale-appropriate sampling methods that can capture the necessary spectral information at the right temporal, spatial and spectral scales.

Data accessibility. The data and R code used in this study are available at https://github.com/annakat/spectral_hypervolume. A preprint of a previous version of this manuscript can be found on bioRxiv at https://doi.org/10.1101/2020.04.24.060483. Spectral metadata can be found on EcoSIS at https://doi.org/10.21232/tYbmucPP and https://doi.org/10.21232/gMpYDoyN.

Authors' contributions. A.K.S.: conceptualization, data curation, formal analysis, investigation, methodology, software, validation, visualization, writing—original draft, writing—review and editing; J.C.-B.: conceptualization, funding acquisition, investigation, methodology, project administration, resources, supervision, writing—original draft, writing—review and editing; S.K.: data curation, investigation, writing—review and editing; P.A.T.: conceptualization, funding acquisition, investigation, methodology, project administration, writing—review and editing; M.D.M.: conceptualization, funding acquisition, project administration, writing—review and editing; J.J.G.: data curation, formal analysis, investigation, writing—review and editing; H.G.: methodology, software, writing—review and editing; R.W.: data curation, formal analysis, methodology, software, writing—review and editing; J.A.G.: conceptualization, data curation, funding acquisition, investigation, methodology, project administration, resources, supervision, writing—original draft, writing—review and editing.

All authors gave final approval for publication and agreed to be held accountable for the work performed therein.

Competing interests. We declare we have no competing interests.

Funding. This work was supported by the National Science Foundation and National Aeronautics and Space Administration through the Dimensions of Biodiversity program (DEB-1342872 grant to J.C.B. and S.E.H., DEB-1342778 grant to P.A.T., DEB-1342827 grant to M.D.M., DEB-1342823 grant to J.A.G.), by the Cedar Creek National Science Foundation Long-Term Ecological Research program (NSF grant no. DEB-1234162), the ASCEND Biology Integration Institute (NSF grant no. DBI 2021898); iCORE/AITF (grant nos. G224150012 and 200700172), NSERC (grant no. RGPIN-2015–05129) and CFI (grant no. 26793) grants to J.A.G.; by the University of Minnesota (Doctoral Dissertation Fellowship to S.K. and J.J.G.), and by University of Minnesota's Department of Ecology, Evolution and Behavior (Summer Research, Crosby, Rothman, Wilkie, Anderson and Dayton funds to J.J.G.). A.K.S. acknowledges the support of the Research Priority Program in Global Change and Biodiversity (URPP GCB) at the University of Zürich.

Acknowledgements. We would like to thank Brett Fredericksen, Ian Carriere, Erin Murdock, Ripley French and Travis Cobb for help with leaf-level sampling. Thanks also to Etienne Laliberté for comments on earlier versions of the manuscript.

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
