## [Peer Review File · Proceedings of the Royal Society B: Biological Sciences]

Review History

RSPB-2020-1364.R0 (Original submission)

Review form: Reviewer 1 (Bill Shipley)

Recommendation

Reject – article is scientifically unsound

Scientific importance: Is the manuscript an original and important contribution to its field?

Good

General interest: Is the paper of sufficient general interest?

Good

Quality of the paper: Is the overall quality of the paper suitable?

Marginal

Is the length of the paper justified?

Yes

Should the paper be seen by a specialist statistical reviewer?

No

Do you have any concerns about statistical analyses in this paper? If so, please specify them explicitly in your report.

Yes

It is a condition of publication that authors make their supporting data, code and materials available - either as supplementary material or hosted in an external repository. Please rate, if applicable, the supporting data on the following criteria.

Is it accessible?

Yes

Is it clear?

Yes

Is it adequate?

Yes

Do you have any ethical concerns with this paper?

No

Comments to the Author

Comments on RSPB-2020-1364

This manuscript is based on a high-quality data set involving long-term experiments on biodiversity of both herbaceous and woody species. Although I have some experience using lab-based NIRS spectrophotometers, I have never used the field-based systems employed here. Nonetheless, the measurements seem to have been done correctly as far as I can tell. Since so many morphological and chemical properties of leaves, known or suspected of being functionally important, can be captured by infrared and visible light spectra, the idea of replacing functional niche measures with spectral niche measures is quite interesting. However, there are also clear knowledge gaps in this transference that were not clearly identified or acknowledged in the paper. First, a trait only makes sense as a niche axis if the trait has functional significance in terms of fitness or demography. While such functional significance has been established for several traits, including some of the traits used here (leaf nitrogen, leaf fibre content, chlorophyll a & b), this is not true of some of the other traits, like some of the pigment concentrations. This problem is increased greatly when we move to the thousands spectral "traits" (i.e. wavelengths). Do the regions of overlap or lack of overlap that are detected (as done in this paper) have any meaning in terms of biotic or abiotic interactions that lie at the heart of niche theory? Although I can see a potential for this to be true, the authors have not convinced me that this is the case. If this is not the case then the inferences about overlap have no meaning in terms of niche theory. Three other general problems that I detected relate to the rather vague hypotheses that are tested, the seeming mixture of different goals, and the use of convex hulls to measure overlap.

H1 is too vague. What does "distinct" mean? There will always be some level of difference, even between different populations of the same species (even between different genotypes) if we look closely enough. Two related problems: (i) overlap and lack of overlap are not really binary options. In a community with S species, there can be overlap with $0, 1, 2, \dots, S-1$ different species. The degree of overlap is important. See (Li et al. 2018, Li and Shipley 2019). The logic and justification for H2 was not clearly presented. Higher spectral niche space is related to greater genotypic and phenotypic variation. So is the logic that greater genotypic variation related to a greater range of environments for which fitness is positive? Something else? Similarly, the logic for H3 was not clear. Why should greater functional complementarity translate to greater productivity? Productivity will mostly relate to resource availability. Why should more resource

availability result in greater functional complementarity? Presumably, the authors are imagining a situation in which resource availability is constant and so more complementarity results in more resource capture?

In lines 99-109, the authors suddenly begin talking about "species identification models" but had not yet described or defined them. Furthermore, while I can understand the practical importance of being able to correctly identify different taxonomic species from their spectral signature, this does not seem to have much to do with niche occupancy and coexistence. This section seems to be an entirely different objective that was tacked on to the paper.

The use of convex hulls to quantify niche overlap is not the best choice because this method (being a multivariate version of range) is quite sensitive to outliers and cannot detect "holes" in niche space. The use of Bonder's hypervolume method is better.

Finally, some of the results seems almost self-evident to me. For instance, we read that "Species' spectral niches were more distinct than their trait-based niches calculated from the 10 chemical and structural foliar traits measured in our study". This seems self-evident, since the spectral niches were based on 1000's of attributes (i.e. wavelengths) while the trait-based niches were based on only 10 attributes. There will be a lot more chances of finding differences with so many more attributes.

In summary, while the data are of high quality, the analysis has some defects that could be corrected, the hypotheses require more work and the logic and evidence linking volume occupancy in spectral space to that in niche space seems weak to me.

Li, Y., and B. Shipley. 2019. Functional niche occupation and species richness in herbaceous plant communities along experimental gradients of stress and disturbance. *Ann Bot* 124:861-867.
 Li, Y., B. Shipley, J. N. Price, V. D. L. Dantas, R. Tamme, M. Westoby, A. Siefert, B. S. Schamp, M. J. Spasojevic, V. Jung, D. C. Laughlin, S. J. Richardson, Y. L. Bagousse-Pinguet, C. Schöb, A. Gazol, H. C. Prentice, N. Gross, J. Overton, M. V. Cianciaruso, F. Louault, C. Kamiyama, T. Nakashizuka, K. Hikosaka, T. Sasaki, M. Katabuchi, C. Frenette Dussault, S. Gaucherand, N. Chen, M. Vandewalle, and M. A. Batalha. 2018. Habitat filtering determines the functional niche occupancy of plant communities worldwide. *Journal of Ecology* 106:1001-1009.

Review form: Reviewer 2

Recommendation

Accept with minor revision (please list in comments)

Scientific importance: Is the manuscript an original and important contribution to its field?

Good

General interest: Is the paper of sufficient general interest?

Good

Quality of the paper: Is the overall quality of the paper suitable?

Good

Is the length of the paper justified?

Yes

Should the paper be seen by a specialist statistical reviewer?

No

Do you have any concerns about statistical analyses in this paper? If so, please specify them explicitly in your report.

No

It is a condition of publication that authors make their supporting data, code and materials available - either as supplementary material or hosted in an external repository. Please rate, if applicable, the supporting data on the following criteria.

Is it accessible?

Yes

Is it clear?

Yes

Is it adequate?

Yes

Do you have any ethical concerns with this paper?

No

Comments to the Author

Niche and fitness differences between species have been documented in many literatures. By defining the plants differentiate in spectral space as a measure of niche differentiation, this study quantified plant niches using hypervolumes delineated by wavelength bands of plant spectra, or 10 functional traits. This is a novel metric approach to calculating niche size and the findings in both experimentally and naturally assembled communities are interesting. I believe that this paper is certainly publishable in this journal.

Specific comments:

Title: More or less, I am concerned about the title "Spectral niches reveal taxonomic identity...".

What is the objective of this study?

1. Background: Should the subtitle be replaced with "Introduction"? This section is generally well written. However, some sentences should be as concise as possible.

2. Methods: The methods seem also comprehensive.

Lines 138-140: a short description on the method to measure these traits?

Line 158: why between 2 and 21, and between 2 and 10?

Line 160: add a reference on z-standardised.

Line 200: Generally we measure the niche of species or population, not individuals

3. Results:

Line 244: Change "Species' niche overlap" into "Species niche overlap"

Line 271: I doubt the term "Individual spectral niche size"

4. Discussion

Line 316: I think that species differentiation in spectral space should be used to distinguish plants niche, rather plant itself.

References: check the format of your references.

Decision letter (RSPB-2020-1364.R0)

17-Aug-2020

Dear Dr Schweiger:

I am writing to inform you that your manuscript RSPB-2020-1364 entitled "Spectral niches reveal taxonomic identity and complementarity in plant communities" has, in its current form, been rejected for publication in Proceedings B. I have received two reviews of your manuscript as well as detailed comments from the associate editor. The reviews, in particular those of reviewer 1, have highlighted out a number of serious critiques with your manuscript as written. However, the associate editor and I feel that, if you are able to adequately address these concerns, that you will have a stronger manuscript for it. Therefore, we are inviting you to submit a substantially revised and re-worked manuscript that thoroughly addresses each of the reviewers' concerns. However, please note that this is not a provisional acceptance.

Sincerely,
 Dr Sarah Brosnan
 Editor, Proceedings B
 mailto: proceedingsb@royalsociety.org

Associate Editor
 Board Member: 1
 Comments to Author:

We have now received two reviews of "Spectral niches reveal taxonomic identity and complementarity in plant communities". The reviewers had very different impressions of this paper as you will see from the comments below. Reviewer 1's has a very serious conceptual critique which needs to be seriously addressed. I will try to divide them into pieces to provide a blueprint for a potential revision that will greatly improve the paper:

1. The title of the paper uses the word "niche" which has perhaps the most fluid and thus confusing definition in ecology. Clearly reviewer 1 and the authors were using different definitions, and in my view the variety of interpretations of the term has rendered it almost useless from the point of view of precise conversation. Reviewer 1 argues "a trait only makes sense as a niche axis if the trait has functional significance in terms of fitness or demography." This seems to be a very different definition of the term compared to what the authors are trying to express. I would argue that this is a problem that arises from the fuzzy definition of the term "niche" itself, and that even defining the term early in the paper is unlikely to prevent readers from using their own (differing) preconceptions about the term while reading the paper. I would

recommend revising the entire manuscript without using the word "niche" and hopefully the language and the paper will become more precise and more consistently interpreted by different readers.

2. Identifying/distinguishing species based on spectra has been achieved by a large number of species, and can't be argued as a major result at this point. In any case, as reviewer 1 notes, this does seem to be a bit of a tangent compared to the other analyses in the paper.

3. Reviewer 1 is also correct about the statistical point: "Species' spectral niches were more distinct than their trait-based niches" that this may be a feature of the higher dimensionality of the dataset. However, this "number of dimensions" effect could be investigated statistically in a number of ways. One approach would be to a priori use 10 wavelengths to compare to 10 traits; another would be to repeatedly use 10 randomly selected wavelengths.

4. As Reviewer 1 argues, the logic of the hypotheses needs to be greatly improved. More clarity in the logic at this point may also imply new analyses.

There is a great deal of potential in this paper, and I hope these comments and reviews are useful with this work moving forward. That said, I expect that addressing Reviewer 1's comment in full will result in an almost entirely new--and much improved--manuscript.

Reviewer(s)' Comments to Author:

Referee: 1

Comments to the Author(s)

Comments on RSPB-2020-1364

This manuscript is based on a high-quality data set involving long-term experiments on biodiversity of both herbaceous and woody species. Although I have some experience using lab-based NIRS spectrophotometers, I have never used the field-based systems employed here.

Nonetheless, the measurements seem to have been done correctly as far as I can tell. Since so many morphological and chemical properties of leaves, known or suspected of being functionally important, can be captured by infrared and visible light spectra, the idea of replacing functional niche measures with spectral niche measures is quite interesting. However, there are also clear knowledge gaps in this transference that were not clearly identified or acknowledged in the paper. First, a trait only makes sense as a niche axis if the trait has functional significance in terms of fitness or demography. While such functional significance has been established for several traits, including some of the traits used here (leaf nitrogen, leaf fibre content, chlorophyll a & b), this is not true of some of the other traits, like some of the pigment concentrations. This problem is increased greatly when we move to the thousands spectral "traits" (i.e. wavelengths).

Do the regions of overlap or lack of overlap that are detected (as done in this paper) have any meaning in terms of biotic or abiotic interactions that lie at the heart of niche theory? Although I can see a potential for this to be true, the authors have not convinced me that this is the case. If this is not the case then the inferences about overlap have no meaning in terms of niche theory. Three other general problems that I detected relate to the rather vague hypotheses that are tested, the seeming mixture of different goals, and the use of convex hulls to measure overlap.

H1 is too vague. What does "distinct" mean? There will always be some level of difference, even between different populations of the same species (even between different genotypes) if we look closely enough. Two related problems: (i) overlap and lack of overlap are not really binary options. In a community with S species, there can be overlap with 0, 1, 2, ..., $S-1$ different species. The degree of overlap is important. See (Li et al. 2018, Li and Shipley 2019). The logic and justification for H2 was not clearly presented. Higher spectral niche space is related to greater genotypic and phenotypic variation. So is the logic that greater genotypic variation related to a greater range of environments for which fitness is positive? Something else? Similarly, the logic for H3 was not clear. Why should greater functional complementarity translate to greater productivity? Productivity will mostly relate to resource availability. Why should more resource

availability result in greater functional complementarity? Presumably, the authors are imagining a situation in which resource availability is constant and so more complementarity results in more resource capture?

In lines 99-109, the authors suddenly begin talking about "species identification models" but had not yet described or defined them. Furthermore, while I can understand the practical importance of being able to correctly identify different taxonomic species from their spectral signature, this does not seem to have much to do with niche occupancy and coexistence. This section seems to be an entirely different objective that was tacked on to the paper.

The use of convex hulls to quantify niche overlap is not the best choice because this method (being a multivariate version of range) is quite sensitive to outliers and cannot detect "holes" in niche space. The use of Bonder's hypervolume method is better.

Finally, some of the results seems almost self-evident to me. For instance, we read that "Species' spectral niches were more distinct than their trait-based niches calculated from the 10 chemical and structural foliar traits measured in our study". This seems self-evident, since the spectral niches were based on 1000's of attributes (i.e. wavelengths) while the trait-based niches were based on only 10 attributes. There will be a lot more chances of finding differences with so many more attributes.

In summary, while the data are of high quality, the analysis has some defects that could be corrected, the hypotheses require more work and the logic and evidence linking volume occupancy in spectral space to that in niche space seems weak to me.

Li, Y., and B. Shipley. 2019. Functional niche occupation and species richness in herbaceous plant communities along experimental gradients of stress and disturbance. *Ann Bot* 124:861-867.
 Li, Y., B. Shipley, J. N. Price, V. D. L. Dantas, R. Tamme, M. Westoby, A. Siefert, B. S. Schamp, M. J. Spasojevic, V. Jung, D. C. Laughlin, S. J. Richardson, Y. L. Bagousse-Pinguet, C. Schöb, A. Gazol, H. C. Prentice, N. Gross, J. Overton, M. V. Cianciaruso, F. Louault, C. Kamiyama, T. Nakashizuka, K. Hikosaka, T. Sasaki, M. Katabuchi, C. Frenette Dussault, S. Gaucherand, N. Chen, M. Vandewalle, and M. A. Batalha. 2018. Habitat filtering determines the functional niche occupancy of plant communities worldwide. *Journal of Ecology* 106:1001-1009.

Referee: 2

Comments to the Author(s)

Niche and fitness differences between species have been documented in many literatures. By defining the plants differentiate in spectral space as a measure of niche differentiation, this study quantified plant niches using hypervolumes delineated by wavelength bands of plant spectra, or 10 functional traits. This is a novel metric approach to calculating niche size and the findings in both experimentally and naturally assembled communities are interesting. I believe that this paper is certainly publishable in this journal.

Specific comments:

Title: More or less, I am concerned about the title "Spectral niches reveal taxonomic identity...". What is the objective of this study?

1. Background: Should the subtitle be replaced with "Introduction"? This section is generally well written. However, some sentences should be as concise as possible.

2. Methods: The methods seem also comprehensive.

Lines 138-140: a short description on the method to measure these traits?

Line 158: why between 2 and 21, and between 2 and 10?

Line 160: add a reference on z-standardised.

Line 200: Generally we measure the niche of species or population, not individuals

3. Results:

Line 244: Change "Species' niche overlap" into "Species niche overlap"

Line 271: I doubt the term "Individual spectral niche size"

4. Discussion

Line 316: I think that species differentiation in spectral space should be used to distinguish plants niche, rather plant itself.

References: check the format of your references.

Author's Response to Decision Letter for (RSPB-2020-1364.R0)

See Appendix A.

RSPB-2021-0366.R0

Review form: Reviewer 3

Recommendation

Major revision is needed (please make suggestions in comments)

Scientific importance: Is the manuscript an original and important contribution to its field?

Excellent

General interest: Is the paper of sufficient general interest?

Good

Quality of the paper: Is the overall quality of the paper suitable?

Acceptable

Is the length of the paper justified?

Yes

Should the paper be seen by a specialist statistical reviewer?

No

Do you have any concerns about statistical analyses in this paper? If so, please specify them explicitly in your report.

Yes

It is a condition of publication that authors make their supporting data, code and materials available - either as supplementary material or hosted in an external repository. Please rate, if applicable, the supporting data on the following criteria.

Is it accessible?

N/A

Is it clear?

N/A

Is it adequate?

N/A

Do you have any ethical concerns with this paper?

No

Comments to the Author

The authors proposed a concept of plant spectral hypervolume (n-dimensional space occupied by plants and delineated by spectral bands) at individual, species and community levels and developed three clear hypotheses linking them to individual growth, species trait hypervolume, and community productivity, respectively. The three hypotheses were tested and supported either in a grassland diversity experiment (BioDIV) or in a tree diversity experiment (FAB) or in both, which highlighted the importance of plant spectral hypervolumes reflecting ecological strategies, community composition and ecosystem function. I really like this idea and the framework developed by the authors, but I still have several major concerns to be considered in revision.

1. Figure 1 showed the comparison of unique hypervolume fraction in spectral space and trait space. The authors claimed that species spectral hypervolumes were more distinct than their trait-based hypervolumes (lines 368-370), because the unique hypervolume fraction reached 90% when including 15 randomly selected spectral bands but did not reach the same level when including all 10 foliar traits (lines 373-377). I cannot agree this result because the author compared unique hypervolume fraction in different dimensions (15 dimension in spectral space vs. 10 dimensions in trait space) and the unique hypervolume fraction should increase logically with the number of dimensions (as shown in Figure 1 too). Therefore, the unique hypervolume fraction should be compared in space of same dimensions. The species spectral hypervolumes were NOT more distinct than their trait-based hypervolumes when they were compared in same dimensions (e.g. space of 10 spectral bands vs. space of 10 foliar traits).

2. Some important information and issues of calculating hypervolumes and performing their set operation (union, intersection, difference) were not given or considered. For example, which function did you use (hypervolume_gaussian or hypervolume_svm?) and how did you set the parameters in the function (e.g. what bandwidth did you choose if use hypervolume_gaussian)? How did you perform set operations for multiple hypervolumes because the function hypervolume_set can only deal with two hypervolumes? Species hypervolumes in spectral and trait space were not only affected by intraspecific variation but also by the number of observations (number of individuals). Therefore the species hypervolumes estimated here with different number of individuals (lines 204-218) are not comparable directly, I would suggest to randomly selected equal number of individuals to estimate species hypervolume. Does individual hypervolumes were estimated with spectral data measured on 3 or 5 leaves for each individual (lines 208-230)? I think the sample size is too small to estimate hypervolume in such a high dimensional space. These issues should be carefully considered and discussed.

3. The results part should be greatly reduced. This part should focus on describing the tables and figures in the main text while the descriptions of tables and figures in Supporting Information that used to help explain the results of figures in main text should be moved to discussions. For example, lines 386-399, 425-432, 433-436, ... If you think some of these descriptions are important and need to be said in results, you may move related table or figure from supporting information to the main text.

Minor comments

Lines 67-68: Replace the symbol “-” in “plants-defined” and “resolution-are” as brackets or commas.

Lines 80-84: The sentence is too long, and be clearer to be separated as two relatively simple ones.

Line 197: A description of IDENT and TreeDivNet is necessary.

Lines 204-208: I would suggest a table to display these information.

Lines 226-227: Why used different instruments for spectral measurement of herbaceous and tree species?

Lines 237-238: What does it mean “interpolating spectra to 1 nm resolution”? Is it related to “covering the wavelength range from 350 nm to 2500 nm in 1024 spectral bands (line 226)”?

Lines 243-245: I am confused whether these foliar traits were measured independently (255-261) or estimated based on leaf spectra (lines 243-249)? It seems all the ten foliar traits were leaf

chemistry, why not consider other life traits (e.g. leaf anatomy, morphology, life history) as you mentioned in line 76 that might be also related to leaf spectra.

Line 281: Reference 32 is not about the hypervolume package.

Line 293: What does "band-wise reflectance" mean? The peak reflectance in that band?

Lines 287-288: Should "fraction of the hypervolume unique to each species" be the ratio of the hypervolume that is occupied by the focal species and not overlapped by any other species to the hypervolume that is occupied by the focal species? Because the unique hypervolume fraction was between 0-1 based on Figure 1.

Figure S2: Perform a decomposition of trait variation into percentage of interspecific trait variation and intraspecific trait variation is a more common and easier way to display the interspecific trait variations and intraspecific trait variations for these traits.

Decision letter (RSPB-2021-0366.R0)

28-Feb-2021

I am writing to inform you that this version of your manuscript RSPB-2021-0366 entitled "Coupling spectral and resource-use complementarity in experimental grassland and forest communities" has, in its current form, been rejected for publication in Proceedings B. This action has been taken on the advice of the reviewer, who has noted three critical points that must be addressed (please see the reviewer's detailed comments, appended below). However, the reviewer, Associate Editor and I all appreciate the goals of your manuscript and the revisions that you have already undertaken. We would be happy to consider a resubmission, provided the comments of the referees are fully addressed. Please note that this is not a provisional acceptance.

The resubmission will be treated as a new manuscript. However, we will approach the same reviewer if this person is available. Please note that resubmissions must be submitted within six months of the date of this email. In exceptional circumstances, extensions may be possible if agreed with the Editorial Office. Manuscripts submitted after this date will be automatically rejected.

Please find below the comments made by the referees, not including confidential reports to the Editor, which I hope you will find useful.

- 1) A 'response to referees' document including details of how you have responded to the comments, and the adjustments you have made.
- 2) A clean copy of the manuscript and one with 'tracked changes' indicating your 'response to referees' comments document.
- 3) Line numbers in your main document.
- 4) Please read our data sharing policies to ensure that you meet our requirements <https://royalsociety.org/journals/authors/author-guidelines/#data>.

Sincerely,
Dr Sarah Brosnan

Editor, Proceedings B
 mailto: proceedingsb@royalsociety.org

Associate Editor Board Member

Comments to Author:

Thanks for your careful revision, which I would argue has greatly improved the manuscript. The reviewer of the latest draft has flagged three very important points, two of which are quite crucial relating to the methods and the third which relates to presentation and directing the reader to the most important results. All of the reviewer's points are quite insightful and addressing them will greatly improve the manuscript.

Reviewer(s)' Comments to Author:

Referee: 3

Comments to the Author(s).

The authors proposed a concept of plant spectral hypervolume (n-dimensional space occupied by plants and delineated by spectral bands) at individual, species and community levels and developed three clear hypotheses linking them to individual growth, species trait hypervolume, and community productivity, respectively. The three hypotheses were tested and supported either in a grassland diversity experiment (BioDIV) or in a tree diversity experiment (FAB) or in both, which highlighted the importance of plant spectral hypervolumes reflecting ecological strategies, community composition and ecosystem function. I really like this idea and the framework developed by the authors, but I still have several major concerns to be considered in revision.

1. Figure 1 showed the comparison of unique hypervolume fraction in spectral space and trait space. The authors claimed that species spectral hypervolumes were more distinct than their trait-based hypervolumes (lines 368-370), because the unique hypervolume fraction reached 90% when including 15 randomly selected spectral bands but did not reach the same level when including all 10 foliar traits (lines 373-377). I cannot agree this result because the author compared unique hypervolume fraction in different dimensions (15 dimension in spectral space vs. 10 dimensions in trait space) and the unique hypervolume fraction should increase logically with the number of dimensions (as shown in Figure 1 too). Therefore, the unique hypervolume fraction should be compared in space of same dimensions. The species spectral hypervolumes were NOT more distinct than their trait-based hypervolumes when they were compared in same dimensions (e.g. space of 10 spectral bands vs. space of 10 foliar traits).

2. Some important information and issues of calculating hypervolumes and performing their set operation (union, intersection, difference) were not given or considered. For example, which function did you use (`hypervolume_gaussian` or `hypervolume_svm`?) and how did you set the parameters in the function (e.g. what bandwidth did you choose if use `hypervolume_gaussian`)? How did you perform set operations for multiple hypervolumes because the function `hypervolume_set` can only deal with two hypervolumes? Species hypervolumes in spectral and trait space were not only affected by intraspecific variation but also by the number of observations (number of individuals). Therefore the species hypervolumes estimated here with different number of individuals (lines 204-218) are not comparable directly, I would suggest to randomly selected equal number of individuals to estimate species hypervolume. Does individual hypervolumes were estimated with spectral data measured on 3 or 5 leaves for each individual (lines 208-230)? I think the sample size is too small to estimate hypervolume in such a high dimensional space. These issues should be carefully considered and discussed.

3. The results part should be greatly reduced. This part should focus on describing the tables and figures in the main text while the descriptions of tables and figures in Supporting Information that used to help explain the results of figures in main text should be moved to discussions. For example, lines 386-399, 425-432, 433-436, ... If you think some of these descriptions are important and need to be said in results, you may move related table or figure from supporting information to the main text.

Minor comments

Lines 67-68: Replace the symbol “-” in “plants-defined” and “esolution-are” as brackets or commas.

Lines 80-84: The sentence is too long, and be clearer to to be separated as two relatively simple ones.

Line 197: A description of IDENT and TreeDivNet is necessary.

Lines 204-208: I would suggest a table to display these information.

Lines 226-227: Why used different instruments for spectral measurement of herbaceous and tree species?

Lines 237-238: What does it mean “interpolating spectra to 1 nm resolution”? Is it related to “covering the wavelength range from 350 nm to 2500 nm in 1024 spectral bands (line 226)”?

Lines 243-245: I am confused whether these foliar traits were measured indpendently (255-261) or estimated based on leaf spectra (lines 243-249)? It seems all the ten foliar traits were leaf chemetry, why not consider other life traits (e.g. leaf anatomy, morphology, life history) as you mentioned in line 76 that might be also related to leaf spectra.

Line 281: Reference 32 is not about the hypervolume package.

Line 293: What does “band-wise reflectance” mean? The peak relectance in that band?

Lines 287-288: Should “fraction of the hypervolume unique to each species” be the ratio of the hypervolume that is occupied by the focal species and not overlapped by any other species to the hypervolume that is occupied by the focal species? Becasuse the unique hypervolume fraction was between 0-1 based on Figure 1.

Figure S2: Perform a decomposition of trait variaiton into percentage of interspecific trait varition and intraspecific trait variaiton is a more common and easier way to display the interspecific trait varitions and intraspecific trait variaitions for these traits.

Author's Response to Decision Letter for (RSPB-2021-0366.R0)

See Appendix B.

RSPB-2021-0568.R0

Review form: Reviewer 3

Recommendation

Major revision is needed (please make suggestions in comments)

Scientific importance: Is the manuscript an original and important contribution to its field?

Excellent

General interest: Is the paper of sufficient general interest?

Good

Quality of the paper: Is the overall quality of the paper suitable?

Good

Is the length of the paper justified?

Yes

Should the paper be seen by a specialist statistical reviewer?

No

Do you have any concerns about statistical analyses in this paper? If so, please specify them explicitly in your report.

Yes

It is a condition of publication that authors make their supporting data, code and materials available - either as supplementary material or hosted in an external repository. Please rate, if applicable, the supporting data on the following criteria.

Is it accessible?

N/A

Is it clear?

N/A

Is it adequate?

N/A

Do you have any ethical concerns with this paper?

No

Comments to the Author

I really appreciate the careful and detailed responses made by the authors, which have addressed all my previous concerns except the first one. I am still not convinced that the unique hypervolume fraction can be compared between spectral and trait space of different dimensions. The authors claimed that "what we aim to illustrate with Fig. 1 is that because spectral data are so high-dimensional, species can occupy more distinct spaces in spectral than in trait space." The spectral data are so high-dimensional because there are many bands, while only ten leaf traits were measured? If more leaf traits (e.g. other leaf chemical components) were measured and the unique hypervolume fraction were calculated in higher dimensional spaces (e.g. up to 20 dimensions as spectral space), the conclusion that species occupy more distinct spaces in spectral than in trait space may no longer hold. In addition, both spectral and trait spaces were not that high dimensional as authors claimed because most of the variations were explained by the first four LD axes (Figure S1). Therefore, calculating and comparing hypervolumes in higher (than 4) dimensional spectral or trait spaces are meaningless, and I would suggest calculating and comparing hypervolumes only at 2-4 dimensional spaces based on LD (or PCA) axes instead of using raw reflectance or trait data.

Decision letter (RSPB-2021-0568.R0)

21-Mar-2021

I am writing to inform you that this version of your manuscript RSPB-2021-0568 entitled "Coupling spectral and resource-use complementarity in experimental grassland and forest communities" has, in its current form, been rejected for publication in Proceedings B. We appreciate your thoughtful revisions on the last version of the manuscript, however the reviewer highlights one remaining area of concern that must be addressed before a final decision can be made (please see the Associate Editor and Reviewer's detailed comments, below). With this in mind we would be happy to consider a resubmission, provided the comments of the referees are fully addressed. However please note that this is not a provisional acceptance.

The resubmission will be treated as a new manuscript. However, we will approach the same reviewers if they are available and it is deemed appropriate to do so by the Editor. Please note

that resubmissions must be submitted within six months of the date of this email. In exceptional circumstances, extensions may be possible if agreed with the Editorial Office. Manuscripts submitted after this date will be automatically rejected.

Please find below the comments made by the referees, not including confidential reports to the Editor, which I hope you will find useful.

- 1) A 'response to referees' document including details of how you have responded to the comments, and the adjustments you have made.
- 2) A clean copy of the manuscript and one with 'tracked changes' indicating your 'response to referees' comments document.
- 3) Line numbers in your main document.
- 4) Please read our data sharing policies to ensure that you meet our requirements <https://royalsociety.org/journals/authors/author-guidelines/#data>.

Sincerely,
Dr Sarah Brosnan
Editor, Proceedings B
<mailto:proceedingsb@royalsociety.org>

Associate Editor Board Member
Comments to Author:

The reviewer found that the manuscript has improved greatly from previous versions. However, there is one rather significant outstanding issue about dimensionality.

I have actually examined the question that the reviewer raises as part of previous work, and on this point the reviewer is point is both insightful and correct. Comparing overlaps of higher versus lower dimensionality should always lead to less overlap in the higher dimensional space, and this should be relatively simple to show this "null" statistical behavior with a few simulations.

That said, the reviewer also suggests a solution which will solve the problem. For the analysis we do not know at present whether the results will be similar after this problem is solved, and in my view this is a major enough re-analysis that we will need to see the more robust analysis suggested by the reviewer before making a final editorial decision.

Reviewer(s)' Comments to Author:

Referee: 3

Comments to the Author(s).

I really appreciate the careful and detailed responses made by the authors, which have addressed all my previous concerns except the first one. I am still not convinced that the unique hypervolume fraction can be compared between spectral and trait space of different dimensions. The authors claimed that "what we aim to illustrate with Fig. 1 is that because spectral data are so high-dimensional, species can occupy more distinct spaces in spectral than in trait space." The spectral data are so high-dimensional because there are many bands, while only ten leaf traits were measured? If more leaf traits (e.g. other leaf chemical components) were measured and the unique hypervolume fraction were calculated in higher dimensional spaces (e.g. up to 20 dimensions as spectral space), the conclusion that species occupy more distinct spaces in

spectral than in trait space may no longer hold. In addition, both spectral and trait spaces were not that high dimensional as authors claimed because most of the variations were explained by the first four LD axes (Figure S1). Therefore, calculating and comparing hypervolumes in higher (than 4) dimensional spectral or trait spaces are meaningless, and I would suggest calculating and comparing hypervolumes only at 2-4 dimensional spaces based on LD (or PCA) axes in stead of using raw reflectance or trait data.

Author's Response to Decision Letter for (RSPB-2021-0568.R0)

See Appendix C.

RSPB-2021-1290.R0

Review form: Reviewer 3

Recommendation

Accept with minor revision (please list in comments)

Scientific importance: Is the manuscript an original and important contribution to its field?

Excellent

General interest: Is the paper of sufficient general interest?

Good

Quality of the paper: Is the overall quality of the paper suitable?

Good

Is the length of the paper justified?

Yes

Should the paper be seen by a specialist statistical reviewer?

No

Do you have any concerns about statistical analyses in this paper? If so, please specify them explicitly in your report.

Yes

It is a condition of publication that authors make their supporting data, code and materials available - either as supplementary material or hosted in an external repository. Please rate, if applicable, the supporting data on the following criteria.

Is it accessible?

Yes

Is it clear?

Yes

Is it adequate?

Yes

Do you have any ethical concerns with this paper?

No

Comments to the Author

It is my third time review this manuscript. I like this paper and I am satisfied with the modifications made in previous version except the comparison of hypervolume fraction between spectral and trait space of different dimensions. In current version, the authors simply removed all parts related to this point. Although the authors did not solve this issue, I agree that left parts are still sufficient to be published. Therefore, I recommend acceptance with the following two points to be checked.

P182-182 The authors choose a quantile threshold of 0%, which will give infinite hypervolume if you use the Gaussian kernel density estimation. The default is 5% in the hypervolume_gaussian function, please check.

P185-187 The authors stated that they randomly selected 9 and 12 individuals per plot to calculate the spectral space occupied by plant communities in FAB and BioDIV, respectively. If the individuals were randomly selected, I did not see if the authors had repeated the random selections many times and use the average hypervolume to represent the spectral space occupied by plant communities rather a just a single selection. If the results were based on a single random selection, they may not be robust. In addition, what is the minimal and maximal number of individuals were sampled measure leaf reflectance per plot, are 9 or 12 individuals sufficient to represent the whole community?

Decision letter (RSPB-2021-1290.R0)

28-Jun-2021

Dear Dr Schweiger:

We have now received comments on your revision by the original reviewer, who notes just a few remaining issues to clarify (these are outlined below in the reviewer and Associate Editor comments). I invite you to revise your manuscript to address them.

Research ethics:

Use of animals and field studies:

It is a condition of publication that you make available the data and research materials supporting the results in the article (<https://royalsociety.org/journals/authors/author-guidelines/#data>). Datasets should be deposited in an appropriate publicly available repository and details of the associated accession number, link or DOI to the datasets must be included in the Data Accessibility section of the article (<https://royalsociety.org/journals/ethics-policies/data-sharing-mining/>). Reference(s) to datasets should also be included in the reference list of the article with DOIs (where available).

Please submit a copy of your revised paper within three weeks. If we do not hear from you within this time your manuscript will be rejected. If you are unable to meet this deadline please let us know as soon as possible, as we may be able to grant a short extension.

Best wishes,
Dr Sarah Brosnan
Editor, Proceedings B
mailto: proceedingsb@royalsociety.org

Associate Editor
Comments to Author:

There is a great deal of progress in this revision. There are a few more issues to deal with related to the Gaussian kernel density estimation. The reviewer is quite knowledgeable about this method and their points are thoughtful and precise in this area. It would also be good to have the code available for future research in this area, since hyperspectral measurements are becoming more common, and there is not currently a consensus for how to analyse those data. The links for the data from ecosis.org also did not work for me, although this could be an issue with the url. Using the DOI would be much more reliable.

Reviewer(s)' Comments to Author:
Referee: 3

Comments to the Author(s).

It is my third time review this manuscript. I like this paper and I am satisfied with the modifications made in previous version except the comparison of hypervolume fraction between spectral and trait space of different dimensions. In current version, the authors simply removed all parts related to this point. Although the authors did not solve this issue, I agree that left parts are still sufficient to be published. Therefore, I recommend acceptance with the following two points to be checked.

P182-182 The authors choose a quantile threshold of 0%, which will give infinite hypervolume if you use the Gaussian kernel density estimation. The default is 5% in the hypervolume_gaussian function, please check.

P185-187 The authors stated that they randomly selected 9 and 12 individuals per plot to calculate the spectral space occupied by plant communities in FAB and BioDIV, respectively. If the individuals were randomly selected, I did not see if the authors had repeated the random selections many times and use the average hypervolume to represent the spectral space occupied by plant communities rather a just a single selection. If the results were based on a single random selection, they may not be robust. In addition, what is the minimal and maximal number of individuals were sampled measure leaf reflectance per plot, are 9 or 12 individuals sufficient to represent the whole community?

Author's Response to Decision Letter for (RSPB-2021-1290.R0)

See Appendix D.

RSPB-2021-1290.R1 (Revision)

Review form: Reviewer 3 (Yuanzhi Li)

Recommendation

Accept as is

Scientific importance: Is the manuscript an original and important contribution to its field?
Good

General interest: Is the paper of sufficient general interest?
Good

Quality of the paper: Is the overall quality of the paper suitable?
Good

Is the length of the paper justified?
Yes

Should the paper be seen by a specialist statistical reviewer?
No

Do you have any concerns about statistical analyses in this paper? If so, please specify them explicitly in your report.
No

It is a condition of publication that authors make their supporting data, code and materials available - either as supplementary material or hosted in an external repository. Please rate, if applicable, the supporting data on the following criteria.

Is it accessible?
Yes

Is it clear?
Yes

Is it adequate?
Yes

Do you have any ethical concerns with this paper?
No

Comments to the Author

I appreciate the efforts of the authors repeating the analyses and am happy to see that the results are robust. I have no more concerns and recommend acceptance.

Decision letter (RSPB-2021-1290.R1)

02-Aug-2021

Dear Dr Schweiger

I am pleased to inform you that your Review manuscript RSPB-2021-1290.R1 entitled "Coupling spectral and resource-use complementarity in experimental grassland and forest communities" has been accepted for publication in Proceedings B.

We require the preprint paper to be referenced. In the Data Accessibility section in our submission system, please add your bioRxiv paper and also add the paper to the reference list.

Because the schedule for publication is very tight, it is a condition of publication that you submit the revised version of your manuscript within 7 days. If you do not think you will be able to meet this date please let me know immediately.

To upload your manuscript, log into <http://mc.manuscriptcentral.com/prsb> and enter your Author Centre, where you will find your manuscript title listed under "Manuscripts with Decisions." Under "Actions," click on "Create a Revision." Your manuscript number has been appended to denote a revision.

You will be unable to make your revisions on the originally submitted version of the manuscript. Instead, upload a new version through your Author Centre.

1) A text file of the manuscript (doc, txt, rtf or tex), including the references, tables (including captions) and figure captions. Please remove any tracked changes from the text before submission. PDF files are not an accepted format for the "Main Document".

2) A separate electronic file of each figure (tiff, EPS or print-quality PDF preferred). The format should be produced directly from original creation package, or original software format. Please note that PowerPoint files are not accepted.

3) Electronic supplementary material: this should be contained in a separate file from the main text and the file name should contain the author's name and journal name, e.g. `authurname_procb_ESM_figures.pdf`

All supplementary materials accompanying an accepted article will be treated as in their final form. They will be published alongside the paper on the journal website and posted on the online figshare repository. Files on figshare will be made available approximately one week before the accompanying article so that the supplementary material can be attributed a unique DOI. Please see: <https://royalsociety.org/journals/authors/author-guidelines/>

4) Data-Sharing and data citation

It is a condition of publication that data supporting your paper are made available. Data should be made available either in the electronic supplementary material or through an appropriate repository. Details of how to access data should be included in your paper. Please see <https://royalsociety.org/journals/ethics-policies/data-sharing-mining/> for more details.

<http://datadryad.org/submit?journalID=RSPB&manu=RSPB-2021-1290.R1> which will take you to your unique entry in the Dryad repository.

Once again, thank you for submitting your manuscript to Proceedings B and I look forward to receiving your final version. If you have any questions at all, please do not hesitate to get in touch.

Sincerely,

Dr Sarah Brosnan

Associate Editor Board Member: 1
Comments to Author:
(There are no comments.)

Board Member: 2
Comments to Author:
(There are no comments.)

Reviewer(s)' Comments to Author:

Referee: 3
Comments to the Author(s)
I appreciate the efforts of the authors repeating the analyses and am happy to see that the results are robust. I have no more concerns and recommend acceptance.

Decision letter (RSPB-2021-1290.R2)

09-Aug-2021

Dear Dr Schweiger

I am pleased to inform you that your manuscript entitled "Coupling spectral and resource-use complementarity in experimental grassland and forest communities" has been accepted for publication in Proceedings B.

Your article has been estimated as being 8 pages long. Our Production Office will be able to confirm the exact length at proof stage.

Data Accessibility section

Open Access

Paper charges

Sincerely,
Editor, Proceedings B
mailto: proceedingsb@royalsociety.org

Appendix A

Associate Editor

Board Member: 1

Comments to Author:

We have now received two reviews of "Spectral niches reveal taxonomic identity and complementarity in plant communities". The reviewers had very different impressions of this paper as you will see from the comments below. Reviewer 1's has a very serious conceptual critique which needs to be seriously addressed. I will try to divide them into pieces to provide a blueprint for a potential revision that will greatly improve the paper:

1. The title of the paper uses the word "niche" which has perhaps the most fluid and thus confusing definition in ecology. Clearly reviewer 1 and the authors were using different definitions, and in my view the variety of interpretations of the term has rendered it almost useless from the point of view of precise conversation. Reviewer 1 argues "a trait only makes sense as a niche axis if the trait has functional significance in terms of fitness or demography." This seems to be a very different definition of the term compared to what the authors are trying to express. I would argue that this is a problem that arises from the fuzzy definition of the term "niche" itself, and that even defining the term early in the paper is unlikely to prevent readers from using their own (differing) preconceptions about the term while reading the paper. I would recommend revising the entire manuscript without using the word "niche" and hopefully the language and the paper will become more precise and more consistently interpreted by different readers.

Thank you very much for this comment. It led to extremely insightful discussions within our group. We have changed the title of our manuscript to "Coupling spectral and resource-use complementarity in experimental grassland and forest communities"; and we have decided to abandon the term "spectral niche" and use "spectral hypervolume" instead for the following reasons:

1) Using the term “spectral niche” is analogous to saying that traits define the niche, when most niche conceptions are about the environment. The niche has traditionally been viewed as the environmental realm a species inhabits (Grinnell, Hutchinson; with Elton’s definition being more food web-centric). Plant functions determine which part of the niche volume organisms can inhabit. But functional traits are not describing the environmental space (the classic “niche” space) itself. We agree that terms and concepts like “niche” are most useful for ecology when they are as precisely defined as possible.

2) Even if we would use “spectral niche” analogously to “trait niche”, we agree with Reviewer 1 that “traits only make sense as a niche axis if they have functional significance in terms of fitness”. Although functional leaf traits, including SLA and nitrogen content, do influence the spectral response, the specific influences of these traits on spectra are difficult to quantify. This is because countless leaf characteristics influence spectral reflectance, many in the form of harmonics of absorption features occurring outside of the spectral range measured, and their effects on plant spectra overlap. In addition, chemical elements, such as nitrogen, are part of a series of macromolecules within leaf tissues, all of which have specific absorption features depending on their chemical composition and conformation. The effects on spectra of these molecules are also influenced by their location within the leaf, i.e., by the number and composition of “boundaries” the incoming light needs to cross before interacting with these molecules. Given these challenges, it is difficult to assign fitness consequences to specific wavelengths or wavelength regions, perhaps with the exception of water and pigment absorption bands.

2. Identifying/distinguishing species based on spectra has been achieved by a large number of species, and can't be argued as a major result at this point. In any case, as reviewer 1 notes, this does seem to be a bit of a tangent compared to the other analyses in the paper.

Yes, we agree. Species differentiation using spectra has been achieved by many studies. However, we think that it is important to show that the degree of species' overlap in spectral space (or lack thereof) provides the basis for the identification of plants with spectra. This result is novel and shows how spectral hypervolumes can explain the established results on spectral species discrimination. We thus would like to keep the analysis showing the contraction of species' hypervolume size with increasing number of spectral axes (figure 1), and the comparison of species differentiation models based on spectra and leaf traits in the BioDIV experiment (figure 2) in the manuscript. However, we agree that the parts of the manuscript dealing with the transferability of spectral models between sites, times of measurement, and levels of observation distract from the focus of the paper (spectral hypervolumes as a means to study species functional differences and the effects of these differences on plant communities). We have removed these parts from the manuscript and are planning to present them in a separate paper.

3. Reviewer 1 is also correct about the statistical point: "Species' spectral niches were more distinct than their trait-based niches" that this may be a feature of the higher dimensionality of the dataset. However, this "number of dimensions" effect could be investigated statistically in a number of ways. One approach would be to a priori use 10 wavelengths to compare to 10 traits; another would be to repeatedly use 10 randomly selected wavelengths.

Yes, we fully agree that the higher dimensionality of the spectral space compared to trait space is the statistical reason why species' spectral hypervolumes overlap less than their hypervolumes in trait space. We added this as a clarification to the introduction (L122) and discussion sections (L344). However, we are also convinced that spectra, because they depend on the chemical and structural composition of leaf tissues, capture more of the total variation in chemical and structural traits than the limited set of foliar traits we (and others) usually measure. We elaborate on this point in the discussion (L 346ff).

In our analysis, comparing species' hypervolume overlap in spectral and traits space, we did, however, select spectral bands at random (see methods L 196ff). We performed 50 random selections of 2 to 21 spectral bands. We are showing up to 21 spectral bands to illustrate the point that the higher dimensionality of the spectral space allows increased separation compared to trait space. We find it remarkable that 15 randomly selected and only 1nm wide bands, out of a total of 2400 bands, led to the hypervolumes of all species being at least 90% unique. This is interesting, because the 10 traits used in our study can be expected to influence the spectral response across a range of wavelength regions. The high degree of separation of species spectral hypervolumes supports the idea of spectra serving as integrated measures of plant phenotypes.

4. As Reviewer 1 argues, the logic of the hypotheses needs to be greatly improved. More clarity in the logic at this point may also imply new analyses.

We agree and are grateful for the suggestions. We have rewritten parts of the manuscript including the hypothesis and think they are much clearer now.

There is a great deal of potential in this paper, and I hope these comments and reviews are useful with this work moving forward. That said, I expect that addressing Reviewer 1's comment in full will result in an almost entirely new--and much improved--manuscript.

Thank you very much for providing us the opportunity to resubmit of our work. Thanks to the reviewers' and your comments, we believe we were able to greatly improved our manuscript. In particular, we think that removing the term "spectral niche" and de-emphasising the species identification models increased the focus of our manuscript and improved its' clarity. We hope that you will be satisfied with our edits and clarifications, and we are looking forward to hearing from you.

Reviewer(s)' Comments to Author:

Referee: 1

Comments to the Author(s)

Comments on RSPB-2020-1364

This manuscript is based on a high-quality data set involving long-term experiments on biodiversity of both herbaceous and woody species. Although I have some experience using lab-based NIRS spectrophotometers, I have never used the field-based systems employed here. Nonetheless, the measurements seem to have been done correctly as far as I can tell. Since so many morphological and chemical properties of leaves, known or suspected of being functionally important, can be captured by infrared and visible light spectra, the idea of replacing functional niche measures with spectral niche measures is quite interesting. However, there are also clear knowledge gaps in this transference that were not clearly identified or acknowledged in the paper. First, a trait only makes sense as a niche axis if the trait has functional significance in terms of fitness or demography. While such functional significance has been established for several traits, including some of the traits used here (leaf nitrogen, leaf fibre content, chlorophyll a & b), this is not true of some of the other traits, like some of the pigment concentrations. This problem is increased greatly when we move to the thousands spectral "traits" (i.e. wavelengths). Do the regions of overlap or lack of overlap that are detected (as done in this paper) have any meaning in terms of biotic or abiotic interactions that lie at the heart of niche theory? Although I can see a potential for this to be true, the authors have not convinced me that this is the case. If this is not the case then the inferences about overlap have no meaning in terms of niche theory.

Thank you very much for your thoughtful comments which led to, so we think, substantial improvements in terms of clarity and focus of our manuscript.

We agree that it would be very difficult to link individual spectral bands to functional traits that have a fitness effect. Linking any leaf trait, functional or not, clearly to specific bands or regions of the reflectance spectrum is complicated by overlapping effects of leaf chemistry, structure and function on spectral reflectance (exceptions include leaf water and pigment content which do have distinct and well-described absorption features). We view spectra of plants as integrated measures of their phenotypes, because many leaf traits (including SLA; water content; the contents of main leaf elements, including C, N, P, Mg; pigments; and more complex molecules, including phenolics and lignin) influence the spectral response and they can be predicted with high accuracies from spectra. Typically, the wavelengths that are most important for predicting those traits occur throughout the spectrum (see e.g., Wang et al. 2018, Serbin et al. 2020). In addition, leaf anatomy, including the number and orientation of cell layers, intercellular spaces and thickness of the cuticle) and morphology, including surface structures such as leaf hairs, influence the spectral response (see e.g., Ustin & Jacquemoud 2020).

We have decided to abandon the term "spectral niche". This is because traditionally the niche space has been described as the environmental realm a species inhabits. We agree with the editor that introducing another definition of the term "niche", which has already one of the most discussed definitions in ecology, distracts from the goals of our work. We aim to show that: 1) The degree to which plant species occupy distinct spectral hypervolumes (i.e. hypervolumes that are unique to the focal species and not overlap by any other species) provides the basis for the spectral differentiation of plant species (and other clades); and that 2) The size and position of spectral hypervolumes occupied by individual plants and plant communities reveal insight into differences in resource strategies, and are, in our case, associated with plant growth and productivity.

Wang, Z., et al., *Mapping foliar functional traits and their uncertainties across three years in a grassland experiment*. Remote sensing of environment, 2019. **221**: p. 405-416.

Serbin, S.P., et al., *From the Arctic to the tropics: multi-biome prediction of leaf mass per area using leaf reflectance*. *New Phytologist*, 2019. **224**(4): p. 1557–1568.

Ustin, S.L. and S. Jacquemoud, *How the Optical Properties of Leaves Modify the Absorption and Scattering of Energy and Enhance Leaf Functionality*, in *Remote Sensing of Plant Biodiversity*. 2020, Springer, Cham. p. 349-384.

Three other general problems that I detected relate to the rather vague hypotheses that are tested, the seeming mixture of different goals, and the use of convex hulls to measure overlap.

We think that one reason for these issues might have been that we did not make clear in the previous version of the manuscript that we are measuring spectral hypervolumes at multiple biological scales, i.e., at the level of plant species, plant individuals and communities. We have added a section to the manuscript describing our expectations for each of these scales (L73ff) and we have rewritten the hypotheses section (L118ff) to clarify.

H1 is too vague. What does "distinct" mean? There will always be some level of difference, even between different populations of the same species (even between different genotypes) if we look closely enough. Two related problems: (i) overlap and lack of overlap are not really binary options. In a community with S species, there can be overlap with 0, 1, 2, ..., $S-1$ different species. The degree of overlap is important. See (Li et al. 2018, Li and Shipley 2019).

Thank you for this comment. We fully agree. For each focal species, out of a number of species, there is often some fraction of the focal species' hypervolume that is unique to the focal species, while other fractions of the hypervolume are overlapped by other species. We have rewritten our hypothesis and hope they are clearer now (L122ff). In addition, we clarify in the methods that with distinct or unique hypervolumes we mean the degree to which a focal species' spectral hypervolume is not overlapped by any other species (L200).

The logic and justification for H2 was not clearly presented. Higher spectral niche space is related to greater genotypic and phenotypic variation. So is the logic that greater genotypic variation related to a greater range of environments for which fitness is positive? Something else?

Thank you for helping us clarify our hypotheses. In this case, we are referring to individual plants and not species. Our logic is that intra-individual foliar trait variation, and thus intra-individual variation in leaf spectra, is linked to plant growth, because plant growth increases canopy complexity, and thus leads to an increasingly diverse light environment experienced by the plant. Thus, we expect spectral hypervolume size occupied by individual plants to be associated with growth. We have rewritten our hypotheses to clarify (L122ff).

Similarly, the logic for H3 was not clear. Why should greater functional complementarity translate to greater productivity? Productivity will mostly relate to resource availability. Why should more resource availability result in greater functional complementarity? Presumably, the authors are imagining a situation in which resource availability is constant and so more complementarity results in more resource capture?

Thank you for pointing out to us that the initial definition of H3 is not without ambiguity. Indeed, we were thinking of situations where resource availability is relatively constant and more complementarity in resource use strategies is expected to increase resource capture and productivity. In the revised version of the manuscript, we keep hypothesis three more general and state that we expect "...that plant communities that occupy greater total spectral space – either due to spectral complementarity among species or due to individuals that occupy large spectral hypervolumes or both – will be associated with more productive ecosystems" (L129).

In lines 99-109, the authors suddenly begin talking about "species identification models" but had not yet described or defined them. Furthermore, while I can understand the practical importance of being able to correctly identify different taxonomic species from their spectral signature, this does not seem to have much to do with niche occupancy and coexistence. This section seems to be an entirely different objective that was tacked on to the paper.

We fully agree that the various species identification models and the investigations of model transferability distract from the focus of our paper – that differences in plant spectral hypervolumes reflect species and community differences that are linked to resource-use complementarity. We removed these parts from the manuscript, keeping only the comparison between species differentiation based on traits and spectra in BioDIV in the paper. We would like to keep this part because i) it substantiates our point that spectra provide an integrated measure of plant phenotypes that makes it easier to differentiate species than a series of commonly measured traits, many of which may be similar among species growing in a common environment, and ii) measurable spectral differences among species are a pre-requisite for spectral hypervolumes to be associated with different resource-use strategies. We believe that removing most of the species identification part greatly improves the focus of the manuscript, and it also allows us to remove the additional datasets used to assess model transferability (old fields chronosequence, oak savanna data). We intend to present these data in a separate manuscript focused on the transferability of species identification models.

The use of convex hulls to quantify niche overlap is not the best choice because this method (being a multivariate version of range) is quite sensitive to outliers and cannot detect “holes” in niche space. The use of Bonder’s hypervolume method is better.

Thank you for catching this mistake. We are indeed using Blonder's hypervolume method. However, it is based on kernel density estimation and not convex hulls. We corrected this in the text (L192ff).

Finally, some of the results seems almost self-evident to me. For instance, we read that "Species 'spectral niches were more distinct than their trait-based niches calculated from the 10 chemical and structural foliar traits measured in our study". This seems self-evident, since the spectral niches were based on 1000's of attributes (i.e. wavelengths) while the trait-based niches were based on only 10 attributes. There will be a lot more chances of finding differences with so many more attributes.

Thank you, we fully agree. It is indeed expected that hypervolumes become more distinct with increasing dimensionality of the hyperspace. We clarify this expectation in the hypotheses section (L122) and discussion (L344ff).

In summary, while the data are of high quality, the analysis has some defects that could be corrected, the hypotheses require more work and the logic and evidence linking volume occupancy in spectral space to that in niche space seems weak to me.

Thank you very much for reviewing our work. Your comments and suggestions have helped us greatly to improve the quality of our manuscript. We hope you are satisfied by our changes and look forward hearing from you.

Li, Y., and B. Shipley. 2019. Functional niche occupation and species richness in herbaceous plant communities along experimental gradients of stress and disturbance. *Ann Bot* 124:861-867.

Li, Y., B. Shipley, J. N. Price, V. D. L. Dantas, R. Tamme, M. Westoby, A. Siefert, B. S. Schamp, M. J. Spasojevic, V. Jung, D. C. Laughlin, S. J. Richardson, Y. L. Bagousse-Pinguet, C. Schöb, A.

Gazol, H. C. Prentice, N. Gross, J. Overton, M. V. Cianciaruso, F. Louault, C. Kamiyama, T. Nakashizuka, K. Hikosaka, T. Sasaki, M. Katabuchi, C. Frenette Dussault, S. Gaucherand, N. Chen, M. Vandewalle, and M. A. Batalha. 2018. Habitat filtering determines the functional niche occupancy of plant communities worldwide. *Journal of Ecology* 106:1001-1009.

Referee: 2

Comments to the Author(s)

Niche and fitness differences between species have been documented in many literatures. By defining the plants differentiate in spectral space as a measure of niche differentiation, this study quantified plant niches using hypervolumes delineated by wavelength bands of plant spectra, or 10 functional traits. This is a novel metric approach to calculating niche size and the findings in both experimentally and naturally assembled communities are interesting. I believe that this paper is certainly publishable in this journal.

Thank you very much for reviewing our manuscript and for the positive evaluation of our work. We have made a number of changes to the manuscript as suggested by the editor and reviewers (see above). Most importantly we are not using the term "spectral niches" anymore, but refer to the spectral space occupied by plants as spectral hypervolumes instead. Moreover, we removed the assessment of the transferability of species identification models from the manuscript and plan to present these results in a separate paper. We think that these changes have improved the clarity of our manuscript and hope that you agree with them. Please see our responses to your questions and suggestions below.

Specific comments:

Title: More or less, I am concerned about the title "Spectral niches reveal taxonomic identity...". What is the objective of this study?

We changed the title of our manuscript to: "Coupling spectral and resource-use complementarity in experimental grassland and forest communities". Our main hypothesis is that spectral complementarity of plants is indicative of their ecological complementarity in terms of resource use (L119). We have in the revised version of the manuscript de-emphasized the species identification models and we have removed the analysis of model transferability, which we aim to present in a separate paper.

1. Background: Should the subtitle be replaced with "Introduction"? This section is generally well written. However, some sentences should be as concise as possible.

Thank you, we agree that "Introduction" works better than "Background". We have rewritten parts of this section and hope the wording has improved.

2. Methods: The methods seem also comprehensive.

Lines 138-140: a short description on the method to measure these traits?

We agree that a short description of laboratory methods is useful for readers and included it in L178ff.

Line 158: why between 2 and 21, and between 2 and 10?

Indeed, we used random selections of between 2 and 21 spectral band (out of 2400) and between 2 and 10 traits (out of 10) to compare the degree of uniqueness of species' hypervolumes in spectral and trait space. The reason for the different number of dimensions is that we expected the degree of uniqueness of the hypervolumes occupied by species to increase with the dimensionality of the space (L122). Statistically, the higher dimensionality of the spectral space compared to trait space is the most likely explanation for species identification models based on spectra outperforming species identification models based

on foliar traits. Biologically, this support the idea that spectra provide integrative measures of plant phenotypes; and that spectra can differentiate species better than a series of commonly measures traits, many of which might be relatively similar among species growing in a common abiotic environment. We discuss these point in L344ff.

Line 160: add a reference on z-standardised.

Perhaps it would be more common to say that variables were "centred and scaled". We use this expression in the revised version of the manuscript (L198). Variables are centred by subtracting the mean and scaled by dividing them by their standard deviation. The advantage of this procedure is that variables with different value ranges are weighted equally in the analysis.

Line 200: Generally, we measure the niche of species or population, not individuals

Thank you, we agree. We changed the wording and now use "the spectral space occupied by individuals" throughout the manuscript.

3. Results:

Line 244: Change "Species 'niche overlap" into "Species niche overlap"

We now use "species hypervolume overlap" throughout the manuscript.

Line 271: I doubt the term "Individual spectral niche size

We changed the terminology throughout the manuscript and now use "hypervolumes" instead of "niches". In this case we changed the wording to "The size of the hypervolume occupied by individual plants is associated with plant growth".

4. Discussion

Line 316: I think that species differentiation in spectral space should be used to distinguish plants niche, rather plant itself.

We agree that this heading was not ideal. It now reads "The degree of species' differentiation in spectral space"

References: check the format of your references.

We have checked and adjusted the reference formatting.

Appendix B

Associate Editor Board Member

Comments to Author:

Thanks for your careful revision, which I would argue has greatly improved the manuscript. The reviewer of the latest draft has flagged three very important points, two of which are quite crucial relating to the methods and the third which relates to presentation and directing the reader to the most important results. All of the reviewer's points are quite insightful and addressing them will greatly improve the manuscript.

Thank you for giving us another change to revise our manuscript. Reviewer 3 did raise some important points regarding our analysis. However, we were able to address all concerns by adding clarifications in the text. In some instances, our description of methods lacked some necessary detail. We are grateful that reviewer 3 has pointed these gaps out to us and have revised the manuscript accordingly. We have also restructured the results and discussion to focus the readers' attention on the results presented in the main part of the manuscript, as suggested. To achieve this, we decided to make Figure 4 a six-panel figure now showing the results for the linkages between the spectral hypervolumes occupied by plant communities in BioDIV and FAB, species richness and productivity in a more succinct way. Thank you for the careful review of our manuscript. We hope that you approve of our changes. We are looking forward to hearing from you.

Reviewer(s)' Comments to Author:

Referee: 3

Comments to the Author(s).

The authors proposed a concept of plant spectral hypervolume (n-dimensional space occupied by plants and delineated by spectral bands) at individual, species and community levels and developed three clear hypotheses linking them to individual growth, species trait hypervolume, and community productivity, respectively. The three hypotheses were tested and supported either in a grassland diversity experiment (BioDIV) or in a tree diversity experiment (FAB) or in both, which highlighted the importance of plant spectral hypervolumes reflecting ecological strategies, community composition and ecosystem function. I really like this idea and the framework developed by the authors, but I still have several major concerns to be considered in revision.

Thank you very much for your encouragement. We have addressed your concerns below and have made edits to the manuscript for clarification. Thanks for helping us making our manuscript better. Please note that our line numbers refer to the document without track changes. Unfortunately, line numbering in MS Word with track changes is inconsistent. We found no easy way to fix this.

1. Figure 1 showed the comparison of unique hypervolume fraction in spectral space and trait space. The authors claimed that species spectral hypervolumes were more distinct than their trait-based hypervolumes (lines 368-370), because the unique hypervolume fraction reached 90% when including 15 randomly selected spectral bands but did not reach the same level when including all 10 foliar traits (lines 373-377). I cannot agree this result because the author compared unique hypervolume fraction in different dimensions (15 dimension in spectral space vs. 10 dimensions in trait space) and the unique hypervolume fraction should increase logically with the number of dimensions (as shown in Figure 1 too). Therefore, the unique hypervolume fraction

should be compared in space of same dimensions. The species spectral hypervolumes were NOT more distinct than their trait-based hypervolumes when they were compared in same dimensions (e.g. space of 10 spectral bands vs. space of 10 foliar traits).

You are absolutely right. At 10 dimensions, there is not a big difference among the degree of species' separation in trait and spectral space. Also, the increase in differentiation among species with increasing dimensionality in both spectral and trait space is to be expected, as you point out. Our claim is that the high dimensionality of spectral data is a strength, allowing it to capture more information about functional differences among plants (and with less measurement time) than most sets of conventionally measured traits. From this perspective, it isn't necessary to compare species separability from the same number of spectral bands and traits, because the advantage of the spectrum is that its many bands capture a much broader range of information useful for discriminating species. We make similar points at various instances in the paper:

“... we expect the high dimensionality of spectral data to detect plant species differences more readily than commonly measured foliar traits, many of which may be similar among species growing in a common abiotic environment.” (Introduction, L123)

“The fraction of the hypervolume space unique to each of our 14 species of grassland–savannah perennials increased with the dimensionality of spectral and trait space” (Results, L266)

“The degree of species differentiation in spectral space is not surprising given the high dimensionality of spectral data.” (Discussion, L343)

Furthermore, we discuss the likely reason for better separability of species in spectral space compared to trait space (L344ff): “...spectra of plants are expected to describe the total dissimilarity among plants resulting from their differences in biochemistry, anatomy and morphology more completely than a number of selected traits. To some extent, this effect could be due to redundancy in our trait measures. In our case, light gradients are probably the dominant source of environmental variation, and all leaf traits measured in our study are to some degree influenced by variation in light.”

Figure 1 illustrates that in spectral space, species can reach a level of uniqueness that they cannot easily reach in trait space. Figure S1 illustrates this point as well; we were not able to find main axes of variation in LDA along which all species separated in trait space (the grasses were especially tricky), while in spectral space we could achieve this separation. This is a reason why species identification models based on spectra outperform species identification based on traits (Fig. 2). We do not doubt that species could be differentiated in trait space to a similar degree as in spectral space if additional traits were used—including flower, seed and root traits, which capture additional differences in terms of resource allocation. However, this would require even more effort in terms of field work than measurements of foliar traits.

It is not at all our intention to disregard the importance of trait measurements. In the contrary, without careful trait measurements, we would not be able to draw any conclusions from spectral data alone. What we aim to illustrate with Fig. 1 is that because spectral data are so high-dimensional, species

can occupy more distinct spaces in spectral than in trait space. Fig. 1 is important to show because if species had shown high degrees of overlap in spectral space, our main hypothesis that the degree of species separation in spectral space provides a measure of resource-use complementarity would have no basis.

2. Some important information and issues of calculating hypervolumes and performing their set operation (union, intersection, difference) were not given or considered. For example, which function did you use (`hypervolume_gaussian` or `hypervolume_svm`?) and how did you set the parameters in the function (e.g. what bandwidth did you choose if use `hypervolume_gaussian`)? How did you perform set operations for multiple hypervolumes because the function `hypervolume_set` can only deal with two hypervolumes?

Thanks for these comments. We clarify in the methods section L196 and L203: “We inferred hypervolumes using Gaussian kernel density estimation and a Silverman bandwidth estimator with a quantile threshold of 0%.” and “Since the `hypervolume_set` function can only deal with two hypervolumes, we compared the spectral and trait hypervolume occupied by a focal species to the spectral and trait hypervolume occupied by all the remaining species.”

Species hypervolumes in spectral and trait space were not only affected by intraspecific variation but also by the number of observations (number of individuals). Therefore the species hypervolumes estimated here with different number of individuals (lines 204-218) are not comparable directly, I would suggest to randomly selected equal number of individuals to estimate species hypervolume.

We agree that with different numbers of observations species hypervolumes are not comparable directly. However, in this case we are not interested in the absolute size of species hypervolumes, but in comparing hypervolumes constructed from the same number of observations in spectral and trait space; and in assessing the degree of overlap between a focal species hypervolume and the hypervolumes of all other species. Since we have the same number of spectral and trait observations for each species, we can make fair comparisons of separability in spectral and trait space.

In the updated version of the ms, we clarify, thanks to your comment above, that we are calculating the degree of overlap for each focal species with all other species by comparing the hypervolume of a focal species (A) to the hypervolume of all other species combined (B). We think that by reducing the number of observations by species, such that A and B contain the same number of observations, we would lose too much information. E.g., when comparing the hypervolume of a species (A) with 40 observations, we could select only 5 observations for each of the other 8 species such that their hypervolume (B) would be calculated based on 40 observations as well.

We think that the richness of our dataset is one of the strengths of the manuscript, especially for calculating the spectral hypervolumes of species and their uniqueness from all other species. We planned data collection in BioDIV to cover the total intraspecific variation in species' traits and spectra within this experiment, sampling across all diversity levels and covering many species combinations. Thus, we think the hypervolumes calculated per species approximate the “true” species' hypervolumes in the experiment quite well. Again, since we are here not interested in

absolute hypervolume size, but in comparing the degree of uniqueness of species hypervolumes' in spectral and trait space, we think that our method is adequate.

For calculating the hypervolumes of individuals and communities, the story is different. In those cases, we are indeed interested in the size of hypervolumes occupied by individuals and species, because we are relating hypervolume size to measures of growth and productivity. Thus, we are using in these cases the same number of spectral measurements per individual (five leaves) and per community (12 randomly selected individuals, L 250). We are sorry that it was not clear that we measured 5 leaves per tree in FAB in the previous version. Less than 5 leaves were only measured for some smaller herbaceous species in BioDIV. We clarify this in the methods L160.

Does individual hypervolumes were estimated with spectral data measured on 3 or 5 leaves for each individual (lines 208-230)? I think the sample size is too small to estimate hypervolume in such a high dimensional space. There issues should be carefully considered and discussed.

Yes, we agree. We used the three main axes of spectral variations calculated from PCA for calculating the spectral hypervolumes of individuals, as we did for plant communities. Somehow this part of the methods section got lost during revisions. Sorry for the oversight. We updated the methods L231ff to clarify. We always measured spectra of 5 leaves per tree in FAB. Less than 5 leaves were only measured for some considerably smaller herbaceous species. We clarify this in the methods L160.

3. The results part should be greatly reduced. This part should focus on describing the tables and figures in the main text while the descriptions of tables and figures in Supporting Information that used to help explain the results of figures in main text should be moved to discussions. For example, lines 386-399, 425-432, 433-436, ... If you think some these description are important and need to be said in results, you may move related table or figure from supporting information to the main text.

Thank you for this suggestion. We have restructured the results and discussion to focus the readers' attention on the main results. We moved discussions of supplementary material to the discussion. In the process, we decided to make Figure 4 a six-panel figure because partitioning the net biodiversity effect in FAB is a key part of our manuscript. Figure 4 now shows the results for the linkages between the spectral hypervolumes occupied by plant communities in BioDIV and FAB, species richness and productivity in a succinct way. These changes have also condensed the main text a bit and shortened the supplement by 2 figures.

Minor comments

Lines 67-68: Replace the symbol “-” in “plants-defined” and “esolution-are” as brackets or commas. We have changed this to commas.

Lines 80-84: The sentence is too long, and be clearer to to be separated as two relatively simple ones.

Thank you for the comment. We separated the sentence into three.

Line 197: A description of IDENT and TreeDivNet is necessary.

We added the description for IDENT and removed TreeDivNet, because it just refers to the platform for sharing IDENT data.

Lines 204-208: I would suggest a table to display these information.

We would like to keep this as is. A table would take up more space and the manuscript is close to the page limit.

Lines 226-227: Why used different instruments for spectral measurement of herbaceous and tree species?

This was because the SVC instrument was not available when we did measurements in FAB. While the SVC and PSR+ are comparable in the range of wavelengths and the spectral resolution they measure, it would require some cross calibration to combine measurements in one analysis. However, in our case it is not a problem that different instruments were used, because we don't mix measurements made by different instruments in any of our analyses.

Lines 237-238: What does it mean "interpolating spectra to 1 nm resolution"? Is it related to "covering the wavelength range from 350 nm to 2500 nm in 1024 spectral bands (line 226)"?

Yes, the native spectral resolution (or spectral sampling intervals) of both instruments varies along the wavelength range (from around 1.4 nm in the visible to 3.8 nm in the near-infrared range). These intervals also change depending on parameter settings such as integration time. Interpolation resamples these intervals to a common interval of 1 nm across all bands, such that wavelengths are set to 400 nm, 401 nm, 402 nm etc. This is a common procedure for making measurements comparable and is done by most instruments internally.

Lines 243-245: I am confused whether these foliar traits were measured independently (255-261) or estimated based on leaf spectra (lines 243-249)? It seems all the ten foliar traits were leaf chemistry, why not consider other life traits (e.g. leaf anatomy, morphology, life history) as you mentioned in line 76 that might be also related to leaf spectra.

We analyzed subsets of around 120 samples per trait (the exact number varies per trait) using the chemical methods mentioned in the methods. These measurements were paired with spectra from the same leaves to develop models for predicting traits from spectral measurements alone. This is a standard procedure that has been used in laboratory NIR analysis for a long time to reduce the amount of samples that need to be chemically processed (which reduces cost and labor). For details see Ref. 15 (including the SI).

We did not measure traits of other organs than leaves, such as root or flower traits, and did not perform any leaf cross-sections to analyze leaf morphology because this paper is less about plant traits, but about using spectra as an additional way to detect and analyze differences in resource use. We aren't trying to figure out whether spectra and traits can discriminate species. We show that spectra can. And we are trying to get insight about *how* they do that, and one way to get that insight is to compare spectral and trait hypervolumes. This comparison reveals that spectra have a lot of information that is useful in discriminating species. We don't want to play out trait measurements against spectral measurements at all, and do not at all intend to argue that one might be better than the other, which would be counterproductive. Functional ecology is the basis for understanding plant spectra. We cannot interpret spectral measurements without thinking about which traits influence specific parts of the spectrum. Thus, we will always need trait and physiological data of plants, as

well as environmental data, to make sense of plant spectra. We try our best to make this clear in the manuscript.

It is a good point that additional trait measurements might increase the discriminatory power of models separating species compared to the ones we are presenting, which are based on chemical traits that are all somewhat related to resource use. We discuss this for instance in L 384ff [many traits of plants cannot be spectrally detected] and L 418ff [redundancy in the leaf traits used in this study]. But again, our focus is on spectral measurements, because they could greatly improve our abilities for monitoring plant species repeatedly across large spatial extents and in areas where it might be difficult to collect trait data on the ground. In this case, we are able to provide insight into *how* spectral models succeed in discriminating species, and can use trait-based models as a heuristic benchmark for performance.

Line 281: Reference 32 is not about the hypervolume package.
Sorry for this. We corrected the reference.

Line 293: What does “band-wise reflectance” mean? The peak reflectance in that band?
It is the reflectance value of each band. At every band between 400-2400 nm the shape of the spectrum is defined by one reflectance value. Our instruments don't provide us with the spectral response function for the original bands measured. Based on the spectral response function, which is Gaussian, the precise peaks of reflectance per band measured could be determined. However, we are confident that the values provided by the instrument capture the spectral response function peak to a good degree. We deleted “band-wise” since it does not contribute anything in this sentence. The sentence now reads “In our case, linear discriminants (LDs) are linear combinations of all reflectance and trait values, respectively, ...”.

Lines 287-288: Should “fraction of the hypervolume unique to each species” be the ratio of the hypervolume that is occupied by the focal species and not overlapped by any other species to the hypervolume that is occupied by the focal species? Because the unique hypervolume fraction was between 0-1 based on Figure 1.

Yes, exactly. Thank you for this comment. We corrected this. The sentence now reads L209: “... [we] calculated the fraction of the hypervolume unique to each species (i.e., the ratio of the hypervolume that is occupied by the focal species and not overlapped by any other species to the hypervolume occupied by the focal species).”

Figure S2: Perform a decomposition of trait variation into percentage of interspecific trait variation and intraspecific trait variation is a more common and easier way to display the interspecific trait variations and intraspecific trait variations for these traits.

Yes, we agree variance partitioning would be a nice way to illustrate this. However, we believe that Figure S3 (Figure S2 in the previous version) is clear enough and serves the purpose of showing that there is considerable intra- and interspecific trait variation in our dataset.

Appendix C

Associate Editor Board Member

Comments to Author:

The reviewer found that the manuscript has improved greatly from previous versions. However, there is one rather significant outstanding issue about dimensionality.

I have actually examined the question that the reviewer raises as part of previous work, and on this point the reviewer is point is both insightful and correct. Comparing overlaps of higher versus lower dimensionality should always lead to less overlap in the higher dimensional space, and this should be relatively simple to show this "null" statistical behavior with a few simulations.

We would like to thank you and the reviewer for the time you have invested in our manuscript, and for providing us with another opportunity to improve our work. It is correct that increasing the number of hypervolume axes always increases the degree of species' separation to some degree. We have attached an R script verifying this statistical behavior even in a null-like case where the axes are randomly generated. One important question is whether or not spectral bands contain information on species identity. If they do, our claim of species occupying more distinct hypervolumes in high-dimensional spectral space compared to less-dimensional trait space would be valid. If they do not, then species' hypervolumes in spectral would not be meaningfully "more distinct" but just smaller and further apart from other species' hypervolumes, as a result of this statistical behavior.

We have looked into the issue of whether or not axes in spectral space contain information on species identity by performing an inclusion (validation) test. We reserved 10 samples per species as validation samples, and we used the rest of the samples for calculating species' hypervolumes in spectral space with increasing number of dimensions. Then, we tested for each species if the validation samples were located within the hypervolume of the same species using the function `hypervolume::hypervolume_inclusion_test` (R package version 2.0.12, Blonder, 2019). We ran 50 iterations for each number of axes (each level of dimensionality), always selecting new band combinations at random. Our result, pasted below (Fig. R1), shows that it becomes less likely for each species to be located within the hypervolume of the same species as the number of axes increases. This means that not all spectral bands contain information on species identity, which increases the chance of samples to be located outside of their respective hypervolumes.

We are convinced that spectra capture more of the total variation in species' chemical and structural traits than the 10 leaf traits we have measured, which is corroborated by our LD analyses and PLSDA models. But we now agree that assessing the degree of hypervolume overlap with increasing dimensionality, as we illustrated in Fig. 1, was not an appropriate way to show this. Most likely, the issue with this analysis is that we selected spectral bands at random, which means that we did not choose specific bands that we suspected in advance would reveal functional differences. This is because we don't yet know which spectral bands are ideal for investigating resource-use complementarity in a particular system. Understanding, at a mechanistic level, which spectral bands are the most useful and why cannot be done with statistics alone, but requires experiments across a wide range of conditions. This is a complex topic which we hope to explore further in the future, but that goes beyond the intended scope of this study.

Fig. R1. Hypervolume inclusion test. The proportion of correctly identified samples per species decreases with the dimensionality of the spectral space.

That said, the reviewer also suggests a solution which will solve the problem. For the analysis we do not know at present whether the results will be similar after this problem is solved, and in my view this is a major enough re-analysis that we will need to see the more robust analysis suggested by the reviewer before making a final editorial decision.

The suggestion of Reviewer 3 to use the first LD axes for delineating species spectral hypervolumes would not address the issue of determining the intrinsic dimensionality of spectral data. We explain this in detail in response to the reviewer's comment below. Briefly, the main axes of spectral variation are dominated by a few leaf characteristics, mainly leaf thickness, chlorophyll, water and dry-matter content. These traits can separate major functional groups (see old Figure S1, pasted in the response to the reviewer's comment). But for distinguishing species it is important that models include minor spectral features that are indicative of subtle differences in chemical and structural traits. This can be seen in the old Figure S1. Only LDs 11-12 were able to differentiate the graminoid species although they each explain less than 1% of the total variation.

Based on this re-assessment of spectral hypervolumes, we have decided to remove Fig. 1 and the associated species identification models from the manuscript. This was already suggested during an earlier review round to make to manuscript more concise. We agree. Now our manuscript focusses on what we think are its most novel and interesting aspects—the coupling of spectral and resource use complementarity, as it is also highlighted in the title. We are convinced that this re-focusing makes are manuscript stronger and we hope that you will support our decision to remove Fig. 1 and the species identification models. We are convinced that our manuscript will spark great interest across research communities and that it will move the uses of spectroscopy for ecosystem function and biodiversity research forward.

Reviewer(s)' Comments to Author:

Referee: 3

Comments to the Author(s).

I really appreciate the careful and detailed responses made by the authors, which have addressed all my previous concerns except the first one. I am still not convinced that the unique hypervolume fraction can be compared between spectral and trait space of different dimensions.

We are pleased that Reviewer 3 is happy with the changes we made to the manuscript and our responses. We have looked into the issue of hypervolume dimensionality. And based on our findings, we agree that unique hypervolume fractions cannot be compared between spectral and trait space of different dimension. We found that the proportion of correctly identified samples per species decreases with the dimensionality of the spectral space. This means that not all spectral bands used in the analysis are informative of species identity—and in addition that, to some degree, overlap among species' hypervolumes in high-dimensional spectral space is reduced by a statistical artefact in which hypervolume overlap decreases with increasing dimensionality independent of information content (see our response above and the R script attached). This statistical artefact is less problematic in trait space because each trait measured is informative about species' differences. Although we are still convinced that spectral data capture species' chemical and structural characteristics more completely than the relatively small sets of leaf traits that are traditionally measured, analyzing the degree of overlap among species' hypervolumes in spaces with increasing dimensionality is not an appropriate way to show this. Most likely, the problem with our approach is that we did not select specific bands that we suspected in advance would reveal functional differences. This is because we don't yet know which spectral bands are ideal for investigating resource-use complementarity in a particular system. Understanding, at a mechanistic level, which spectral bands are the most useful and why cannot be done with statistics alone, but requires experiments across a wide range of conditions. This is a complex topic that should be investigated but goes beyond the scope of this study. For now, we have decided to remove this part of the analyses and the associated species identification models from the manuscript. We are convinced that re-focusing our manuscript on the coupling of spectral and resource use complementarity, as promised in the title, makes a much stronger and clearer contribution. Thanks for giving us the opportunity to investigate the dimensionality issue in more detail.

The authors claimed that “what we aim to illustrate with Fig. 1 is that because spectral data are so high-dimensional, species can occupy more distinct spaces in spectral than in trait space.” The spectral data are so high-dimensional because there are many bands, while only ten leaf traits were measured? If more leaf traits (e.g. other leaf chemical components) were measured and the unique hypervolume fraction were calculated in higher dimensional spaces (e.g. up 20 dimensions as spectral space), the conclusion that species occupy more distinct spaces in spectral than in trait space may no longer hold.

Yes, we agree. We would expect the degree of uniqueness of species' hypervolumes to increase—particularly if non-leaf traits were included, as we discussed in L 356 ff. However, since we have removed the comparison of species hypervolumes in spectral and trait space from the manuscript, we also removed this part of from the discussion.

In addition, both spectral and trait spaces were not that high dimensional as authors claimed because most of the variations were explained by the first four LD axes (Figure S1). Therefore, calculating and comparing hypervolumes in higher (than 4) dimensional spectral or trait spaces are

meaningless, and I would suggest calculating and comparing hypervolumes only at 2-4 dimensional spaces based on LD (or PCA) axes instead of using raw reflectance or trait data.

We appreciate the idea, but we disagree with this suggestion. Subtle spectral features which explain little of the total variance in spectral data play a critical role in spectroscopic methods, including estimating leaf traits from spectra (see e.g., variable importance plots in Serbin et al. 2014, Wang et al. 2018) and species identification (Durgante et al. 2013). The main axes of variation of fresh leaf spectra are dominated by a few leaf characteristics, including leaf thickness, chlorophyll, water and dry-matter content. However, we know that many more leaf traits, including their entire chemical composition (accessory pigments, cell wall constituents, phenolics, tannins, etc.), as well as the structure of the leaf surface, epidermis, and mesophyll layers influence the spectral response (Ustin and Jacquemoud 2020). Thus, although a few axes explain the majority of variance in spectral data, minor spectral features play an important role in distinguishing species even though they explain only small amounts of the total variance. This can be seen in the old Figure S1 (pasted below). While the main axes of spectral variation (LD1-2) separated major functional groups, only LDs 11-12 are able to differentiate the graminoid species although they each explain less than 1% of the total variation. Thus, the dimensionality (*sensu* “information content”) of spectral data cannot be assumed based on the number of ordination axes that explain the main portion of variance. This is because there are certain aspects of spectra that occupy minor fractions of the total variance that are important for differentiating species (especially closely related ones), as they represent subtle absorption features caused by specific leaf structural and chemical components.

We think that what could reveal the “information content” of spectral data are investigations directed at understanding, at a mechanistic level, which spectral bands are the most useful for revealing plant functional differences in particular systems. These insights cannot be gained with statistics alone, but require experiments across a wide range of conditions. However, although we would like to investigate this in the future, it is a complex topic that goes beyond the intended scope of this study.

We would like to thank you for the thoughtfulness with which you have treated the dimensionality issue. We hope that you will support our decision of removing Fig. 1 and the associated species identification models. This re-focuses our manuscript on its strongest aspects, the coupling of spectral and resource complementary among individuals and within communities. Thank you again for handling our manuscript.

Old Figure S1. Species clustering along linear discriminant (LD) axes maximising the differences among species based on (a-c) spectra and (d-f) foliar traits. The amount of the total variation explained by each LD axis is shown in parentheses.

Literature

- Blonder, B. with contributions from Harris, D. J. hypervolume: High Dimensional Geometry and Set Operations Using Kernel Density Estimation, Support Vector Machines, and Convex Hulls. R package version 2.0.12. <https://CRAN.R-project.org/package=hypervolume> (2019).
- Serbin, S. P., Singh, A., McNeil, B. E., Kingdon, C. C. & Townsend, P. A. Spectroscopic determination of leaf morphological and biochemical traits for northern temperate and boreal tree species. *Ecological Applications* **24**, 1651-1669 (2014).
- Wang, Z. *et al.* Mapping foliar functional traits and their uncertainties across three years in a grassland experiment. *Remote sensing of environment* **221**, 405-416 (2019).
- Durgante, F. M., Higuchi, N., Almeida, A. & Vicentini, A. Species spectral signature: discriminating closely related plant species in the Amazon with near-infrared leaf-spectroscopy. *Forest Ecology and Management* **291**, 240-248 (2013).

Appendix D

Associate Editor

Comments to Author:

There is a great deal of progress in this revision. There are a few more issues to deal with related to the Gaussian kernel density estimation. The reviewer is quite knowledgeable about this method and their points are thoughtful and precise in this area. It would also be good to have the code available for future research in this area, since hyperspectral measurements are becoming more common, and there is not currently a consensus for how to analyse those data. The links for the data from ecosis.org also did not work for me, although this could be an issue with the url. Using the DOI would be much more reliable.

We are very happy to hear that you and Referee 3 are overall satisfied with our revision. We agree with Referee 3 that iterating our random selection procedure of spectra to characterize communities is important for testing if results are robust. We re-ran our analyses using an iteration procedure as suggested, and we are happy to report that results remained remarkably stable. So much so that including the standard deviations of hypervolume sizes in Figure 3 did not make sense, they were hardly visible. We thus decided to include the range of standard deviations and hypervolume sizes in the results section (b) and use the average hypervolume sizes for plotting.

Sorry for the data link not working. We have updated it. All data used in this study and the R code can now also be found on Github: https://github.com/annakat/spectral_hypervolumes.

Please note that while going through the data, we noticed four mis-labelled individuals in FAB. Since we could not recover the identity of these measurements, we have excluded them from further analysis. This change from 532 to 524 individuals changed the results for calculations of individual spectral hypervolumes in FAB only minimally (Figure 2).

Thank you very much for handling our manuscript!

Reviewer(s)' Comments to Author:

Referee: 3

Comments to the Author(s).

It is my third time review this manuscript. I like this paper and I am satisfied with the modifications made in previous version except the comparison of hypervolume fraction between spectral and trait space of different dimensions. In current version, the authors simply removed all parts related to this point. Although the authors did not solve this issue, I agree that left parts are still sufficient to be published. Therefore, I recommend acceptance with the following two points to be checked.

We are delighted to hear that you are overall satisfied with our revisions. We reran our analyses iterating the randomization procedure as suggested, and are happy to report that results remained stable (see below). We really appreciated your thoughtful comments and suggestions throughout the review process. Thanks for helping us improve our manuscript!

P182-182 The authors choose a quantile threshold of 0%, which will give infinite hypervolume if you use the Gaussian kernel density estimation. The default is 5% in the `hypervolume_gaussian` function, please check.

Thank you for catching this mistake. Indeed, we used the default settings for `hypervolume_gaussian` with a 5% quantile threshold. We have corrected this in the text.

P185-187 The authors stated that they randomly selected 9 and 12 individuals per plot to calculate the spectral space occupied by plant communities in FAB and BioDIV, respectively. If the individuals were randomly selected, I did not see if the authors had repeated the random selections many times and use the average hypervolume to represent the spectral space occupied by plant communities rather a just a single selection. If the results were based on a single random selection, they may not be robust. In addition, what is the minimal and maximal number of individuals were sampled measure leaf reflectance per plot, are 9 or 12 individuals sufficient to represent the whole community?

Thank you for pointing out these potential issues to us. First, we'd like to answer how many individuals were sampled in the two experiments. Sorry for not including more details on the sampling scheme in the methods. We wanted to keep them simple but we agree, it is important to be clear on the number of individuals measured per plot. We have updated the methods, sections (b) and (d), accordingly. We also

noticed that for FAB we confused the number of measurements with the number of individuals in L202. We have corrected this mistake.

In our sampling design, we balanced the number of individuals sampled per plot against the total number of plots sampled. We decided to maximize the number of plots and sample the number of individuals per plot depending on species richness. In FAB, we measured 3 individuals per species at 3 different height levels: two measurements were taken at the top canopy layer, two at the mid- and one at the bottom-canopy layer. Including measurement at different heights was critical for characterizing individual trees spectrally, and for calculating individual spectral hypervolumes. Measurements per height level were averaged. For FAB, this sampling scheme resulted in 9 spectra from 3 individuals in monocultures, 18 spectra from 6 individuals in bi-cultures, 45 spectra from 15 individuals in 5-species plots, and 108 spectra from 36 individuals in 12-species plots. Some mismeasurements (resulting from lamp failure, a not fully closed leaf clip and other sampling mistakes) were excluded from analysis. For calculating the hypervolumes occupied by plant communities in FAB, we randomly selected 9 spectra per plot, since this allowed us to include the monocultures which had the fewest measurements in our analysis. We repeated the random selection process 50 times (see below).

In BioDIV, we divided each plot into 9 equally sized subplots and collected spectral data in 4 to 8 subplots, depending on species richness. The centre subplot was always excluded to prevent disturbance. In mono- and bi-cultures, we sampled the 4 corner subplots; in 4-species plots, we sampled the 4 corner subplots and 2 additional outer subplots; in 8- or 16-species plots, we sampled all 8 outer subplots. We characterised the plant community in BioDIV by measuring 4 randomly selected individuals of the most abundant species per subplot making sure all species per plot were included, for more details see methods and SI in Schweiger et al. 2018 (doi:10.1038/s41559-018-0551-1). In BioDIV, most plant individuals were small (under 30 cm) showing less intra-individual spectra variability than the trees in FAB. We thus averaged spectra per individual in BioDIV. This sampling scheme resulted in 16 individuals measured in mono- and bi-cultures, 24 individuals measured in 4-species plots, and 32 individuals measured in 8- or 16-species plots. For calculating the hypervolumes occupied by plant communities in BioDIV, we randomly selected 4 subplots per plot and 3 individuals per subplot resulting in 12 individuals per plot. We used 3 random individuals per subplot because we excluded mismeasurements which reduced the sample size, and 4 subplots to allow mono- and bi-cultures to be included. Again, we repeated the random selection process 50 times (see below).

Thanks also for pointing out the importance of iterating our randomization procedure. We reran the calculation of community hypervolumes in FAB and BioDIV using 50 iterations each. The size of community hypervolumes was consistent in both FAB and BioDIV. In FAB, $\log(\text{hypervolume})$ occupied by plant communities ranged from 1.04 to 4.54 and with standard deviations between 0.0054 and 0.0083. In BioDIV, $\log(\text{hypervolume})$ occupied by plant communities ranged from 0.48 to 3.39 and with standard deviations between 0.0051 and 0.0072. We added a description of the iteration procedure (L 204ff), the range of values and standard deviations to the manuscript. Based on the consistency (small standard deviations) of these results we are convinced that our results are stable and that our selection procedure described the hypervolume occupied by the communities adequately. All data and code can be found on Github: https://github.com/annakat/spectral_hypervolumes.

Thanks again very much for your time and effort!